# Human LAMP1 accelerates Lassa virus fusion and potently promotes fusion pore dilation upon forcing viral fusion with non-endosomal membrane

You Zhang[1], Juan Carlos de la Torre[2], Gregory B. Melikyan[1,3]*

1 Department of Pediatrics, Division of Infectious Diseases Emory University School of Medicine, Atlanta, Georgia, United States of America, 2 Department of Immunology and Microbiology, The Scripps Research Institute, La Jolla, California, United States of America, 3 Children's Healthcare of Atlanta, Atlanta, Georgia, United States of America

* gmeliki@emory.edu

**Data Availability Statement:** All relevant data are within the manuscript and its Supporting Information files.

## Abstract

Lassa virus (LASV) cell entry is mediated by the interaction of the virus glycoprotein complex (GPC) with alpha-dystroglycan at the cell surface followed by binding to LAMP1 in late endosomes. However, LAMP1 is not absolutely required for LASV fusion, as this virus can infect LAMP1-deficient cells. Here, we used LASV GPC pseudoviruses, LASV virus-like particles and recombinant lymphocytic choriomeningitis virus expressing LASV GPC to investigate the role of human LAMP1 (hLAMP1) in LASV fusion with human and avian cells expressing a LAMP1 ortholog that does not support LASV entry. We employed a combination of single virus imaging and virus population-based fusion and infectivity assays to dissect the hLAMP1 requirement for initiation and completion of LASV fusion that culminates in the release of viral ribonucleoprotein into the cytoplasm. Unexpectedly, ectopic expression of hLAMP1 accelerated the kinetics of small fusion pore formation, but only modestly increased productive LASV fusion and infection of human and avian cells. To assess the effects of hLAMP1 in the absence of requisite endosomal host factors, we forced LASV fusion with the plasma membrane by applying low pH. Unlike the conventional LASV entry pathway, ectopic hLAMP1 expression dramatically promoted the initial and full dilation of pores formed through forced fusion at the plasma membrane. We further show that, while the soluble hLAMP1 ectodomain accelerates the kinetics of nascent pore formation, it fails to promote efficient pore dilation, suggesting the hLAMP1 transmembrane domain is involved in this late stage of LASV fusion. These findings reveal a previously unappreciated role of hLAMP1 in promoting dilation of LASV fusion pores, which is difficult to ascertain for endosomal fusion where several co-factors, such as bis(monoacylglycero)phosphate, likely regulate LASV entry.

**Funding:** This work was supported by the NIH/
NIAID R01 AI053668 grant to GBM. The funders
had no role in study design, data collection and
analysis, decision to publish, or preparation of the
manuscript.

**Competing interests:** The authors have declared
that no competing interests exist.

## Author summary

Lassa virus (LASV) enters cells *via* fusion with acidic endosomes mediated by the viral gly-
coprotein complex (GPC) interaction with the intracellular receptor LAMP1. However,
the requirement for LAMP1 is not absolute, as LASV can infect avian cells expressing a
LAMP1 ortholog that does not interact with GPC. To delineate the role of LAMP1 in
LASV entry, we developed assays to monitor the formation of nascent fusion pores, as
well as their initial and complete dilation to sizes that allow productive infection of avian
cells by LASV GPC pseudoviruses. This novel approach provided unprecedented details
regarding the dynamics of LASV fusion pores and revealed that ectopic expression of
human LAMP1 in avian cells leads to a marked acceleration of fusion but modestly
increases the likelihood of complete pore dilation and infection. In contrast, human
LAMP1 expression dramatically enhanced the propensity of nascent pores to fully enlarge
when LASV fusion with the plasma membrane was forced by exposure to low pH. Thus,
whereas the role of LAMP1 in LASV fusion is confounded by an interplay between multi-
ple endosomal factors, the plasma membrane is a suitable target for mechanistic dissection
of the roles of host factors in LASV entry.

## Introduction

LASV is an Old World mammarenavirus that infects a broad host range of cells from different
species. LASV cell entry is mediated by the viral surface glycoprotein complex (GPC), a tri-
meric class I fusion protein that consists of non-covalently associated surface (GP1) and trans-
membrane (GP2) subunits (reviewed in [1,2]). The GP1 and GP2 subunits are generated
through cleavage of the GPC precursor by the host cell protease subtilisin kexin isozyme-1
(SKI-1)/site 1 protease (S1P). GP1 is involved in receptor binding and GP2 in membrane
fusion. A unique feature of arenavirus GPC proteins is that their stable signal peptide (SSP),
which is cleaved off the GP precursor, remains associated with the GPC and plays an impor-
tant regulatory role in low pH-induced conformational changes in GPC that lead to fusion of
the viral and cell membranes [3–8]. LASV GPC attachment to the alpha-dystroglycan receptor
on the cell surface leads to virus internalization and transport to acidic endosomes where low
pH promotes virus dissociation from alpha-dystroglycan and attachment to the intracellular
receptor, LAMP1. LAMP1, a marker for late endosomes/lysosomes, has been shown to serve
as a specific receptor for LASV and not for other mammarenaviruses, such as lymphocytic
choriomeningitis virus (LCMV) [9]. Human LAMP1 (hLAMP1), but not human LAMP2 or
avian LAMP1, promotes LASV fusion with late endosomes [9].

Recent studies by others and our group have shown that hLAMP1, while promoting LASV
fusion and infection, is not absolutely required for virus entry, since cells lacking human
LAMP1 support basal levels of LASV fusion/infection [10–12]. The ability of LASV GPC to
mediate membrane fusion in the absence of hLAMP1 is consistent with the reports that suffi-
ciently acidic pH induces irreversible GPC conformational changes leading to shedding of the
GP1 subunit [9,12,13]. Mechanistic studies revealed that hLAMP1 binding shifts the pH-opti-
mum for GPC-mediated fusion to a higher pH [10,12]. Thus, LASV fusion with hLAMP1
expressing cells is likely initiated in less acidic maturing endosomal compartments, prior to
virus delivery into late endosomes/lysosomes.

Our recent study identified a novel co-factor required for completion of LASV fusion–the
late-endosome-resident lipid, bis(monoacylglycero)phosphate (BMP) [12]. BMP specifically
and potently promotes the late stages of LASV GPC-mediated fusion–formation and

enlargement of fusion pores. Whereas hLAMP1 binding clearly augments low pH-dependent refolding of LASV GPC, whether this intracellular receptor also modulates late steps of virus fusion remains unclear. Using a cell-cell fusion model, we have shown that hLAMP1 overexpression facilitates transition from hemifusion (merger of contacting membrane leaflets without fusion pore formation) to full fusion [12]. However, the role of hLAMP1 in controlling distinct steps of virus-endosome fusion has not been elucidated.

Here, we employed a combination of single LASV pseudovirus (LASVpp) tracking in live cells alongside bulk virus-cell fusion and infectivity assays to assess the effect of hLAMP1 on distinct steps of viral fusion. Pseudoviruses co-labeled with viral content marker and an internal pH sensor enable detection of single virus fusion events and monitoring the initial enlargement of fusion pores, whereas bulk fusion/infectivity assay report functional enlargement of fusion pores. We find that hLAMP1 overexpression in both human and avian cell lines moderately promotes fusion and infection through an endocytic entry pathway, whereas LASV GPC-mediated fusion at the cell surface forced by exposure to low pH was dramatically enhanced and accelerated in hLAMP1-expressing cells. Real-time imaging of forced LASVpp fusion with avian cells revealed a strong enhancement in fusion pore dilation by upon hLAMP1 expression. Our results thus provide new insights into the role of hLAMP1 in early and late stages of LASV fusion and reveal key differences in permissiveness of endosomes and the plasma membrane for LASV fusion.

## Results

### Ectopic expression of hLAMP1 potentiates LASV fusion and infection of human and avian cells

We used two cell lines to examine the role of LAMP1 in LASV fusion–human lung epithelial A549 cells endogenously expressing human LAMP1 (hLAMP1) and chicken DF-1 fibroblasts expressing the avian LAMP1 ortholog which does not support LASV entry [9]. Immunofluorescence staining confirmed that A549 cells expressed a basal level of hLAMP1, in agreement with our immunoblotting data [12], whereas this protein was not detectable in DF-1 cells (Fig 1). We next stably transduced A549 and DF-1 cells with wild-type hLAMP1 (LAMP1-WT) or with the LAMP1-d384 mutant which lacks an endocytic signal and is primarily expressed on the cell surface [9]. Transduction with hLAMP1 yielded robust levels of expression in both A549 and DF-1 cells, with hLAMP1 primarily distributed to intracellular compartments (Fig 1A–1D). To assess the levels of hLAMP1 on the cell surface, immunofluorescence staining was performed without cell membrane permeabilization. This protocol revealed modest levels of LAMP1-WT on the surface of both A549 and DF-1 cell lines overexpressing LAMP1-WT, whereas the surface levels of LAMP1-d384 were much higher, reaching approximately half of the total expressed hLAMP1 (Fig 1A–1D).

Consistent with the requirement for hLAMP1, infection of DF-1 cells by HIV-1 particles pseudotyped with LASV GPC (LASVpp) measured by a single-cycle infectivity assay was ~100 fold lower than infection of A549 cells (Fig 2A). To rule out the presence of species-specific endosomal restriction factors in DF-1 cells, we bypassed endosomal entry of LASVpp by triggering virus fusion with the plasma membrane through low pH exposure (referred to as forced fusion, see Materials and Methods for details). Prior to adding the virus and triggering fusion by acidic pH, cells were pre-treated with Bafilomycin A1 (BafA1) to raise endosomal pH and block the endosomal entry of LASVpp. The forced fusion protocol resulted in ~10-fold less efficient LASVpp infection in both DF-1 and A549 cells compared to endosomal entry (Fig 2A). Thus, DF-1 cells are poorly permissive for LASVpp infection and forced LASV GPC-

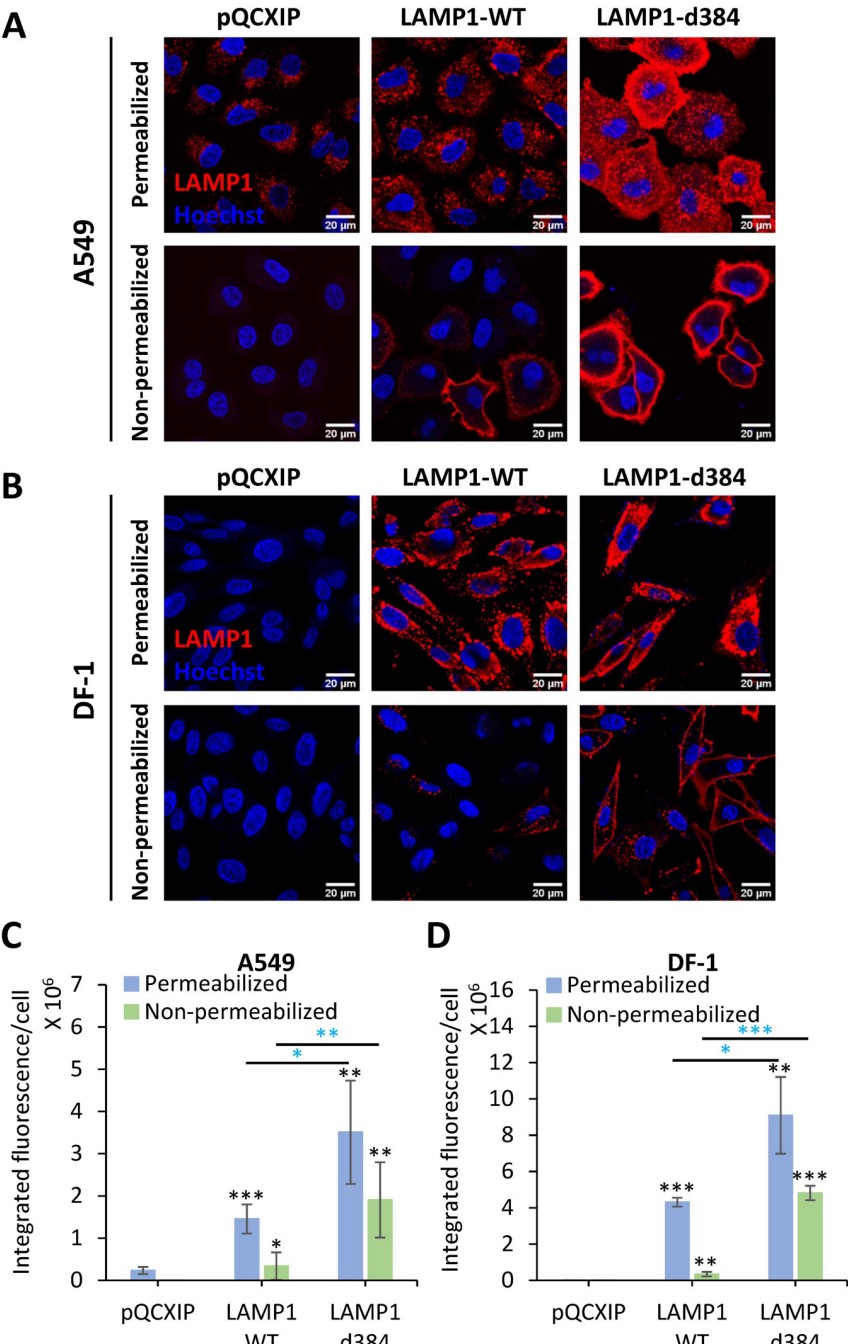

**Fig 1. Analysis of endogenous and ectopic expression and localization of hLAMP1 in A549 and DF-1 cells.** A549 or DF-1 cells were transduced with an empty pQCXIP vector (control) or vector expressing human LAMP1-WT or the LAMP1-d384 mutant. Cells were fixed and whether left untreated (bottom panels in A and B) or permeabilized with 125 μg/ml digitonin and immunostained for human LAMP1. **(A, C)** Images and quantification of hLAMP1 expression in A549 cells. Images acquired under same exposure conditions, but the brightness and contrast settings in panel A and B are different to ensure optimal display. **(B, D)** Images and quantification of hLAMP1 expression with DF-1 cells. Data shown are means ± SD of five fields of view for each condition. Blue asterisks show significance levels for the difference between LAMP1-WT and LAMP1-d384. Black asterisks on the top of bars represent significance relative to the vector control. Data were analyzed by Student's t-test. *, p<0.05; **, p<0.01; ***, p<0.001.

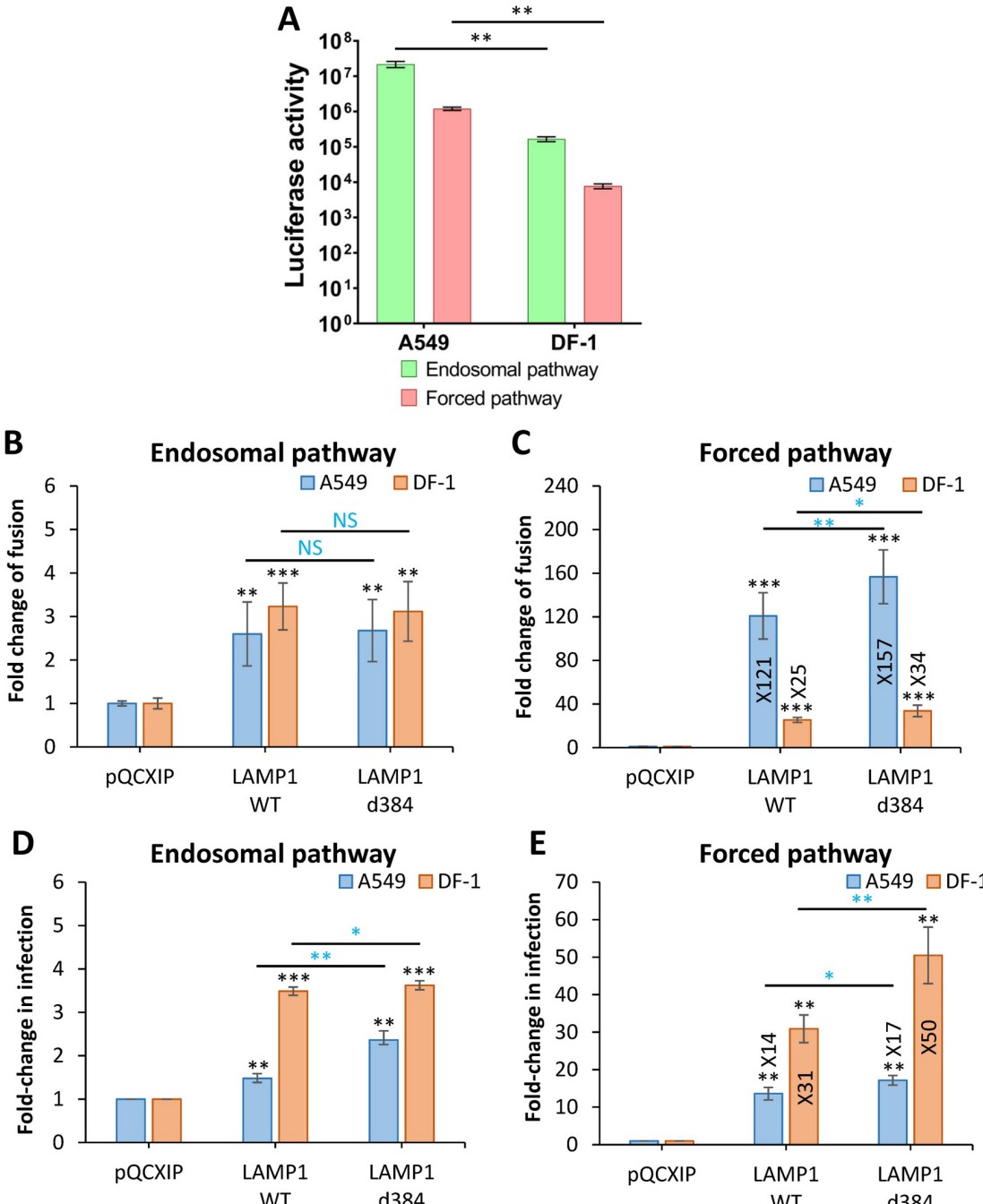

**Fig 2. LAMP1 expression enhances LASVpp fusion and infection in A549 and DF-1 cells.** **(A)** Efficiency of A549 and DF-1 cell infection through endosomal and forced fusion protocols, using luciferase-encoding LASVpp. Infection was measured at 48 hours post-infection (see Methods). **(B)** LASVpp-BlaM fusion with A549 and DF-1 cells. LASVpp entry through an endosomal pathway was initiated by pre-binding pseudoviruses in the cold, shifting to 37˚C and incubating for 2 h. **(C)** Low pH-forced fusion of LASVpp with A549 and DF-1 cells. Cells were pretreated with 0.2 μM BafA1 for 1 h prior to binding pseudoviruses in the cold. Fusion was triggered by applying pH 5.0 citrate buffer at 37˚C for 20 min followed by additional incubation in a neutral pH medium at 37˚C for 30 min. **(D)** LASVpp infection of A549 and DF-1 cells. Luciferase-encoding LASVpp were bound to cells in cold. Cells were incubated in 37˚C for 36 h to allow infection. **(E)** LASVpp infection through low pH bypass protocol in A549 and DF-1 cells described in panel C. After forced fusion, cells at 37˚C for 36 h before reading the resulting luciferase signal. Data shown are means ± SD of three independent experiments. Data were analyzed by Student's t-test. *, p<0.05; **, p<0.01; ***, p<0.001; NS, not significant. Blue asterisks show the significance levels for the difference between LAMP1-WT and LAMP1-d384, black asterisks on the top of bars represent significance relative to the vector control.

mediated fusion with the plasma membrane results in less efficient infection of human or avian cells compared to a conventional entry route.

Next, we examined the effect of ectopic hLAMP1 expression on the efficiency of LASVpp fusion with A549 and DF-1 cells using a direct virus-cell fusion assay that reports the delivery of virus-incorporated beta-lactamase (BlaM) into the cytoplasm [14,15]. DF-1 or A549 cells were infected with pseudoviruses containing BlaM-Vpr chimera (S1A Fig) and viral fusion was measured by the cytoplasmic BlaM activity. Ectopic expression of LAMP1-WT or LAMP1-d384 resulted in a modest, ~3-fold, increase in virus fusion and single-cycle infection of both A549 and DF-1 cells, although hLAMP1 expression somewhat more strongly enhanced infection of DF-1 cells compared to A549 cells (Fig 2B and 2D, for raw data, see S2A and S2C Fig).

A modest increase in LASVpp fusion and infection upon hLAMP1 expression indicates that endogenous levels of this receptor in A549 cells may be sufficient for near-optimal LASVpp entry. In contrast, the lack of a strong effect on viral fusion and infection of DF-1 cells was unexpected, suggesting that endosomal factors other than LAMP1 may facilitate LASV fusion in DF-1 cells.

## hLAMP1 expression dramatically promotes forced LASV pseudovirus fusion with the plasma membrane

To probe the effects of hLAMP1 on LASVpp fusion with membranes that do not contain significant amounts of this intracellular receptor or other endosomal factors that may facilitate LASV entry and to ensure a more tractable system for viral fusion, we bypassed the internalization and endosomal trafficking steps by forcing LASVpp fusion with the plasma membrane through exposure to low pH. Ectopic expression of hLAMP1 dramatically enhanced the forced LASVpp fusion with both A549 and DF-1 cells, as measured by the BlaM assay (Figs 2C and S2B). Expression of LAMP1-WT or LAMP1-d384 in DF-1 cells caused a somewhat less dramatic (~30-fold) increase in LASVpp fusion compared to A549 cells in which ~120-160-fold increase in signal was detected. The much stronger effect of hLAMP1 expression on forced LASVpp fusion with A549 compared to DF-1 cells (Fig 2C) was somewhat unexpected, given that the former cells express endogenous hLAMP1, small amounts of which may be present at the cell surface. The striking increase in the efficiency of forced LASVpp fusion is in agreement with the marked enhancement of LASV GPC-mediated cell-cell fusion upon ectopic expression of hLAMP1 in DF-1 cells [12]. Consistent with the higher expression of the mutant LAMP1 on the cell surface (Fig 1), forced fusion with both A549 and DF-1 cells expressing LAMP1-d384 was significantly more efficient than fusion with cells expressing LAMP1-WT (Figs 2C and S2B). Thus, ectopic hLAMP1 expression greatly enhances the otherwise sub-optimal low pH-mediated LASVpp fusion with the plasma membrane, in contrast to fusion with endosomes which is less affected by hLAMP1 overexpression.

Akin to the effect of hLAMP1 overexpression on the forced viral fusion (Fig 2C), forced LASVpp infection was also potently enhanced by ectopic expression of this receptor in both cell types (Figs 2E and S2D). hLAMP1 expression had a commensurate effect on viral fusion and infection in DF-1 cells, whereas an increase in fusion efficiency was much more pronounced compared to infection in A549 cells (Fig 2C and 2E). A discordance between fold-enhancement of forced fusion *vs* infection upon hLAMP1 expression in A549 cells might be due to less efficient post-fusion steps of infection following LASVpp entry at the plasma membrane compared to entry from endosomes in these but not DF-1 cells.

Given the strikingly potent enhancing effect of ectopic hLAMP1 expression on the forced LASVpp fusion, we sought to determine if such enhancement may be caused by a global, non-

specific effect of hLAMP1 on the plasma membrane. Toward this goal, we measured the fusion of control pseudoviruses bearing the unrelated VSV G protein (VSVpp) with the parental and hLAMP1 expressing cells. As expected, ectopic expression of hLAMP1 did not considerably enhance VSVpp fusion through an endosomal or forced pathway (S3A and S3B Fig). This finding rules out a non-specific effect of hLAMP1 on VSV G-mediate virus fusion with endosomes or the plasma membrane.

## hLAMP1 enhances fusion and infection of LASV VLPs and recombinant Lassa viruses

In order to validate the results obtained with HIV-1-based LASVpp, we measured the fusion of arenavirus virus-like particles (VLPs) bearing LASV GPC (referred to as LASV-VLP). VLPs were made by co-expressing the New World Junin arenavirus NP and Z proteins with LASV GPC, essentially as described in [16]. To measure VLP-cell fusion, we constructed a JUNV NP-beta-lactamase chimera which efficiently incorporated into LASV-VLPs (S1 Fig). Akin to the results for LASVpp fusion (Fig 2B), LASV-VLP fusion with A549 and DF-1 cells measured by the NP-BlaM delivery into the cytoplasm was modestly augmented by ectopic expression of LAMP1-WT or LAMP1-d384 (S4A and S4C Fig). Also, similar to LASVpp, forced LASV-VLP fusion with DF-1 and A549 cells expressing LAMP1-WT or LAMP1-d384 was dramatically (20-40-fold) enhanced compared to control cells (S4B and S4D Fig). Of note, the forced LASV-VLP fusion with A549 cells was less markedly increased upon hLAMP1 expression than forced fusion of LASVpp (compare Figs 2C and S4B, S4D). These results further demonstrate that GPC-mediated fusion with the plasma membrane is much more dependent on ectopic hLAMP1 expression than fusion with endosomes.

Finally, we examined the effect of hLAMP1 expression on infection by recombinant LCMV viruses bearing LASV GPC (LCMV-LASV GPC) [17,18]. A549 and DF-1 cells transduced with LAMP1-WT, LAMP1-d384 or an empty vector were inoculated with LCMV-LASV GPC, and infection was quantified by microscopy after immunostaining for the LCMV NP protein (Fig 3A). Consistent with our results with LASVpp infection (Fig 2D), ectopic expression of hLAMP1 modestly increased recombinant LASV infection in both cell lines (Fig 3B). However, unlike the forced LASVpp and LASV-VLP fusion and infection, which were dramatically enhanced in cells overexpressing hLAMP1 (Figs 2C and 2E and S2B), forced LCMV-LASV GPC infection increased only ~6-fold in DF-1 and A549 cells (Fig 3C). Thus, while the magnitude of hLAMP1 effects on fusion/infection of pseudovirus, VLP and recombinant arenavirus varies, forced GPC-mediated fusion at the plasma membrane is consistently more dependent on ectopic hLAMP1 expression than fusion through a conventional endosomal entry pathway.

## Single LASV pseudovirus fusion proceeds through a viral membrane permeabilization step, irrespective of cell type or hLAMP1 expression levels

We next sought to determine which steps of viral fusion are facilitated by hLAMP1 expression by single virus tracking. LASVpp were labeled with the mCherry-2xCL-YFP-Vpr construct, which is incorporated into HIV-1 pseudoviruses and is cleaved by the viral protease upon virus maturation, producing free mCherry and YFP-Vpr [19]. Loss of mCherry signal from the viral particle entering a cell reflects mCherry release into the cytoplasm through a fusion pore, whereas YFP-Vpr is retained in the viral core for a considerable time and thus serves as a reference marker for reliable detection of single fusion events. Also, importantly, the pH-sensitive YFP fluorescence is quenched at low pH [20], thereby reporting changes in intraviral pH. Using this marker, we have previously shown that LASVpp fusion with endosomes is preceded by a drop in intraviral pH due to permeabilization of the viral membrane in acidic endosomes

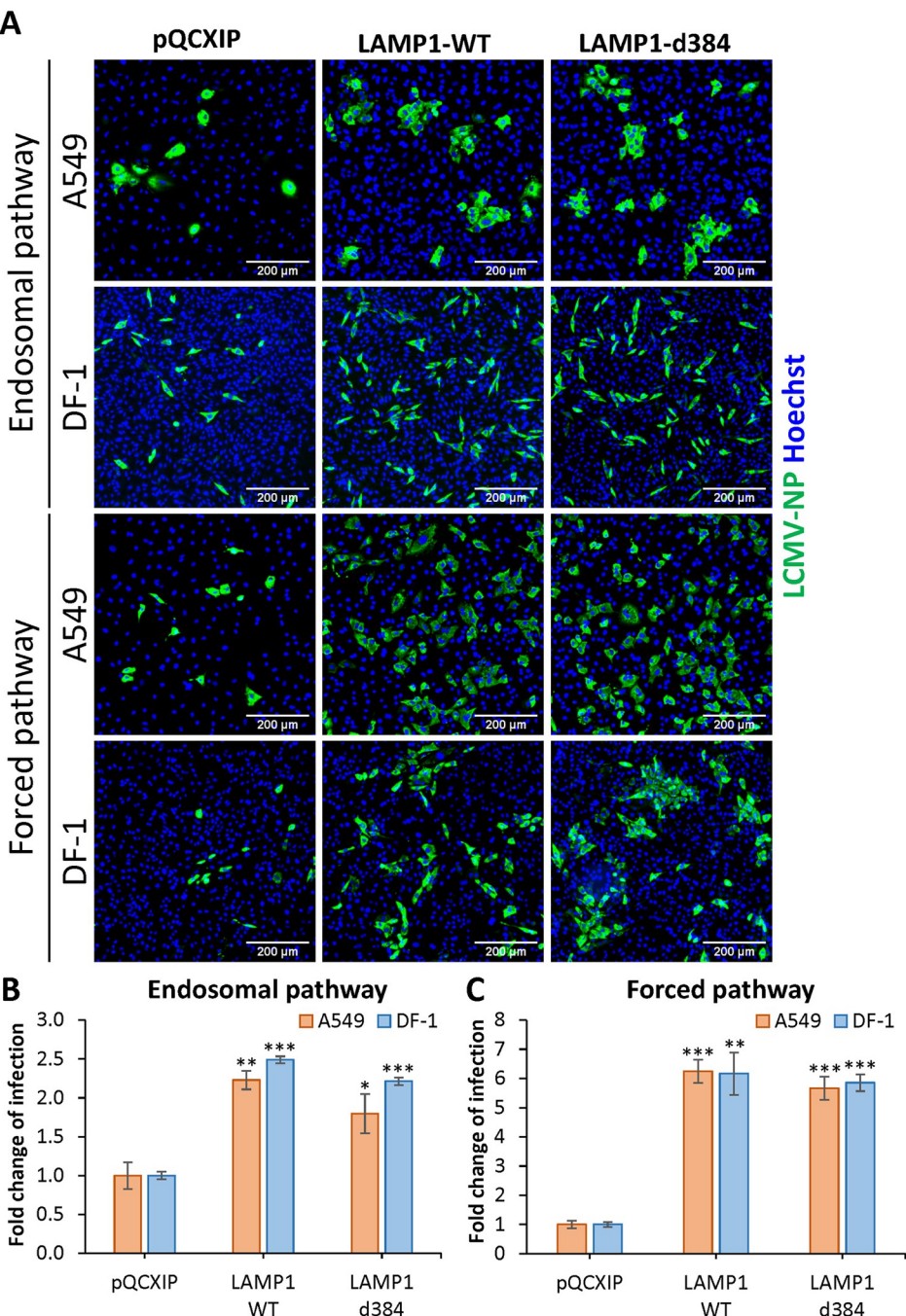

**Fig 3. LAMP1 expression enhances recombinant LCMV/LASV-GPC infection of A549 and DF-1 cells. (A)** Recombinant LCMV/LASV-GPC infection in A549 and DF-1 cells. Cells were allowed to bind LCMV/LASV-GPC in the cold and subsequently incubated at 37°C for 36 h. For infection in A549 and DF-1 cells achieved through forcing viral fusion by low pH exposure, cells were pretreated with 0.2 μM BafA1 for 1 h prior to binding LCMV/LASV-GPC in the cold. Fusion was triggered by applying pH 5.0 citrate buffer at 37°C for 20 min followed by additional incubation in a neutral pH medium at 37°C for 36 h. Infection was detected by immunostaining for LCMV nucleoprotein (NP). **(B)** Quantification of the fold change of the infection through an endosomal pathway. **(C)** Quantification of the fold change of the infection through forced fusion. Data shown are means ± SD of three independent experiments. Data were analyzed by Student's t-test. *, $p < 0.05$; **, $p < 0.01$; ***, $p < 0.001$.

[20]. This loss of a barrier function of the viral membrane prior to fusion is manifested in YFP fluorescence (green) quenching, whereas subsequent virus fusion results in simultaneous loss of mCherry (red) and recovery of YFP signal due to re-neutralization of the viral interior connected to the cytoplasm through a fusion pore (Fig 4A). We refer to fusion events preceded by viral membrane permeabilization as type II fusion, whereas fusion occurring without a prior loss of the viral membrane barrier function are termed type I fusion events [20]. Our pilot results suggest that viral membrane permeabilization in type II fusion events is caused by conformational changes in LASV GPC and requires virus-cell contact prior to exposure to low pH. The above double labeling of pseudoviruses with the content marker, mCherry, and the viral core-associated pH sensor, YFP-Vpr, enables sensitive detection of nascent proton-permeable fusion pores, as well as their initial dilation that allows mCherry escape into the cytoplasm.

Using the aforementioned virus labeling strategy, we imaged single LASVpp fusion with DF-1 endosomes of cells expressing or lacking hLAMP1. LASVpp undergoes type II fusion with A549 cells ([20]). We also found all type II fusion events with control DF-1 cells (S5 Fig). DF-1 cells were chosen over A549 cells for the subsequent imaging experiments for the following reasons: (1) low permissiveness to LASV fusion/infection in the absence of ectopically expressed hLAMP1; (2) consistent effects of hLAMP1 expression on fusion and infection across different virus platforms (LASVpp, LASV-VLP and LCMV-LASV GPC) (Figs 2 and 3 and S5 Fig); and (3) tolerance to prolonged exposure to low pH, which is essential for the 1 hour-long forced fusion imaging experiments described below. All single LASVpp fusion events in DF-1 cells were of type II phenotype, regardless of the hLAMP1 expression (Figs 4B and 4C and S5). Interestingly, a fraction of single LASVpp fusion events exhibited a delayed mCherry release relative to YFP dequenching, which marks the opening of a nascent fusion pore (Fig 4C). This lag in mCherry release ranged from several seconds to minutes (see below) and was most likely caused by delayed enlargement of nascent fusion pores to sizes (~4 nm) that allowed mCherry release. Thus, YFP dequenching in the context of type II virus-endosome fusion provides a highly sensitive means to detect very small fusion pores, whereas mCherry release is contingent on pore enlargement to a diameter exceeding ~4 nm.

In addition to type II fusion events culminating in mCherry loss, we also observed YFP quenching and subsequent dequenching without mCherry release for as long as we tracked viral particles (Fig 4D). The lack of mCherry release could be due to a failure of nascent fusion pores to enlarge and allow mCherry release or due to a full fusion of immature viral particles in which the mCherry marker was not cleaved off the Vpr-based core marker by the HIV-1 protease [19]. Indeed, almost 20% of particles contained uncleaved mCherry-YFP-Vpr construct, as judged by the lack of mCherry release upon saponin lysis *in vitro* (S6 Fig). However, assuming that HIV-1 maturation does not affect GPC-mediated fusion, the fraction of "no-release" events should be constant across conditions. So, the greater fraction of "instantly" dilating pores, but not pores that failed to release mCherry upon ectopic hLAMP1 expression (see below), argues against the possibility that all "no-release" events correspond to fusion of immature particles. Another reason for YFP dequenching without mCherry release could be virus recycling to the cell surface. However, events that did not culminate in mCherry release were also observed upon low pH-forced virus fusion with the plasma membrane where low pH was maintained throughout the experiment (see below), suggesting that virus cycling to the cell surface is less likely to be responsible for YFP dequenching. We therefore conditionally refer to YFP dequenching without mCherry release as stalled fusion.

Of note, all three types of fusion pores–quickly and slowly dilating and those that fail to release mCherry–were observed in A549 cells (S7A–S7D Fig). Regardless of the timing of mCherry release relative to YFP dequenching, LASVpp fusion was associated with a mixed

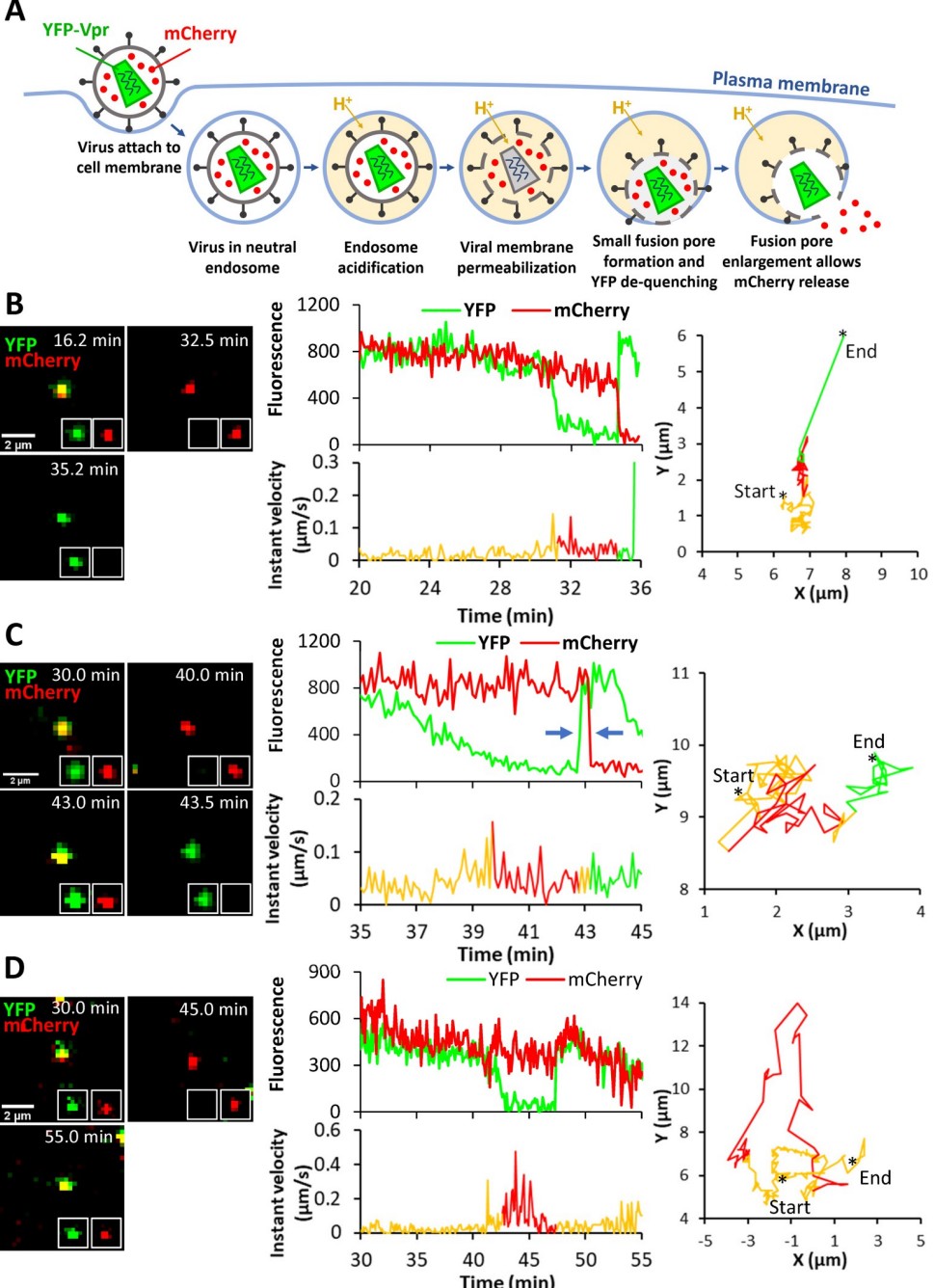

**Fig 4. Single LASVpp fusion with DF-1 cells. (A)** Illustration of fusion of mCherry-2xCL-YFP-Vpr labeled single LASVpp in an acidic endosome. An increase in the virus membrane permeability leads to quenching of the intraviral YFP signal (green) in acidic environment. Subsequent virus fusion with the endosomal membrane results in a loss of mCherry signal (red) through a fusion pore and concomitant re-neutralization of virus interior, as evidenced by YFP signal dequenching. **(B)** LASVpp fusion events (YFP dequenching) with instant mCherry release (quick fusion pore dilation). Time lapse images (left), fluorescence traces (middle top), instant velocity (middle bottom) and trajectory (right) of single LASVpp fusion with DF-1-LAMP1-WT cell showing YFP quenching at 31.3 min and YFP dequenching/mCherry loss at 34.7 min corresponding to virus interior acidification and fusion, respectively (see S1 Movie). **(C)** LASVpp fusion events with delayed mCherry release relative to YFP dequenching. Time lapse images (left), fluorescence traces (middle top), instant velocity (middle bottom) and trajectory (right) of single LASVpp fusion with a DF-1-LAMP1-WT cell showing YFP quenching at 39.7 min, YFP dequenching at 42.7 min and mCherry loss at 43.2 min (arrows), indicating virus interior acidification, small fusion pore formation and fusion pore dilation to a diameter exceeding the size of mCherry, respectively (see S2 Movie). **(D)** LASVpp fusion events (YFP dequenching)

without mCherry release. Time lapse images (left), fluorescence traces (middle top), instant velocity (middle bottom) and trajectory (right) of single LASVpp fusion with a DF-1-LAMP1-WT cell showing YFP quenching at 42.6 min and dequenching at 47.4 min without mCherry loss (see S3 Movie).

diffusive and directional motion pattern (Fig 4, right graphs), which is typical for endosomal trafficking of internalized cargo and viruses (e.g., [21]).

## hLAMP1 overexpression accelerates single LASV pseudovirus fusion with endosomes without affecting dilation of fusion pores

Analysis of the kinetics of nascent fusion pore formation (YFP dequenching) revealed a ~2-fold increase in the fusion rate with DF-1 cells expressing LAMP1-WT or LAMP1-d384 compared to control cells (Fig 5A). The faster kinetic of single LASVpp fusion with hLAMP1-expressing DF-1 cells was not caused by more efficient or faster virus endocytosis and delivery into acidic endosomes. Neither the fraction of particles exhibiting YFP quenching (i.e., virus interior acidification in acidic endosomes irrespective of subsequent fusion) nor the kinetics of YFP quenching were dependent of hLAMP1 expression (S8A and S8B Fig). Notably, only ~15% of cell-bound particles exhibited YFP quenching in acidic endosomes of DF-1 cells expressing or lacking hLAMP1, suggesting that LASVpp uptake was slow/inefficient. Of note, the kinetics of nascent LASVpp pore formation in parental A549 cells was much faster than in DF-1 cells and as fast as the accelerated kinetics of hLAMP1 expressing DF-1 cells (compare Figs 5A to S7E), highlighting the cell type-dependence of the multi-step LASVpp entry process.

Importantly, hLAMP1 expression markedly shortened the lag between virus interior acidification (YFP quenching) and small pore formation (YFP dequenching) from ~18 min to ~8 min, on average (Fig 5B). The larger lag to small fusion pore formation after virus entry into acidic compartments may thus be responsible for the slower LASVpp fusion in cells lacking hLAMP1 (Fig 5A).

In DF-1 cells, we observed three single LASVpp fusion phenotypes–instant and delayed pore enlargement and stalled fusion,–irrespective of ectopic hLAMP1 expression (Fig 5C). All three types of single fusion events were promoted ~4-6-fold upon hLAMP1 expression, in general agreement with the bulk LASVpp fusion results (Fig 2B). It should be stressed, however, that hLAMP1 expression did not significantly alter the relative weights of different types of fusion events, including the "instantly" dilating pores (Fig 5C, *Inset*). Another approach to evaluate the effect of hLAMP1 on the propensity of fusion pores to enlarge is to assess the time required for nascent pore (detected by YFP dequenching) to dilate to sizes that allow mCherry release. "Instant" pore enlargement was defined as simultaneous (within our temporal resolution of 6 sec) YFP dequenching and mCherry loss. The lag time to mCherry release was not significantly affected by hLAMP1 expression (Fig 5D), in excellent agreement with the finding that hLAMP1 did not significantly increase the fraction of "instant" mCherry release events (Fig 5C, *Inset*). These results show that ectopic expression of hLAMP1 does not noticeably promote the initial enlargement of fusion pores formed between LASVpp and endosomes.

## hLAMP1 expression accelerates forced fusion of single LASV pseudoviruses at the cell surface and promotes initial dilation of fusion pores

We and others have previously shown that hLAMP1 overexpression allows LASV GPC-mediated fusion to occur at higher pH [10,12]. To control the pH that triggers LASV GPC conformational changes, we employed the forced fusion protocol (Fig 6A). Fluorescent LASVpp

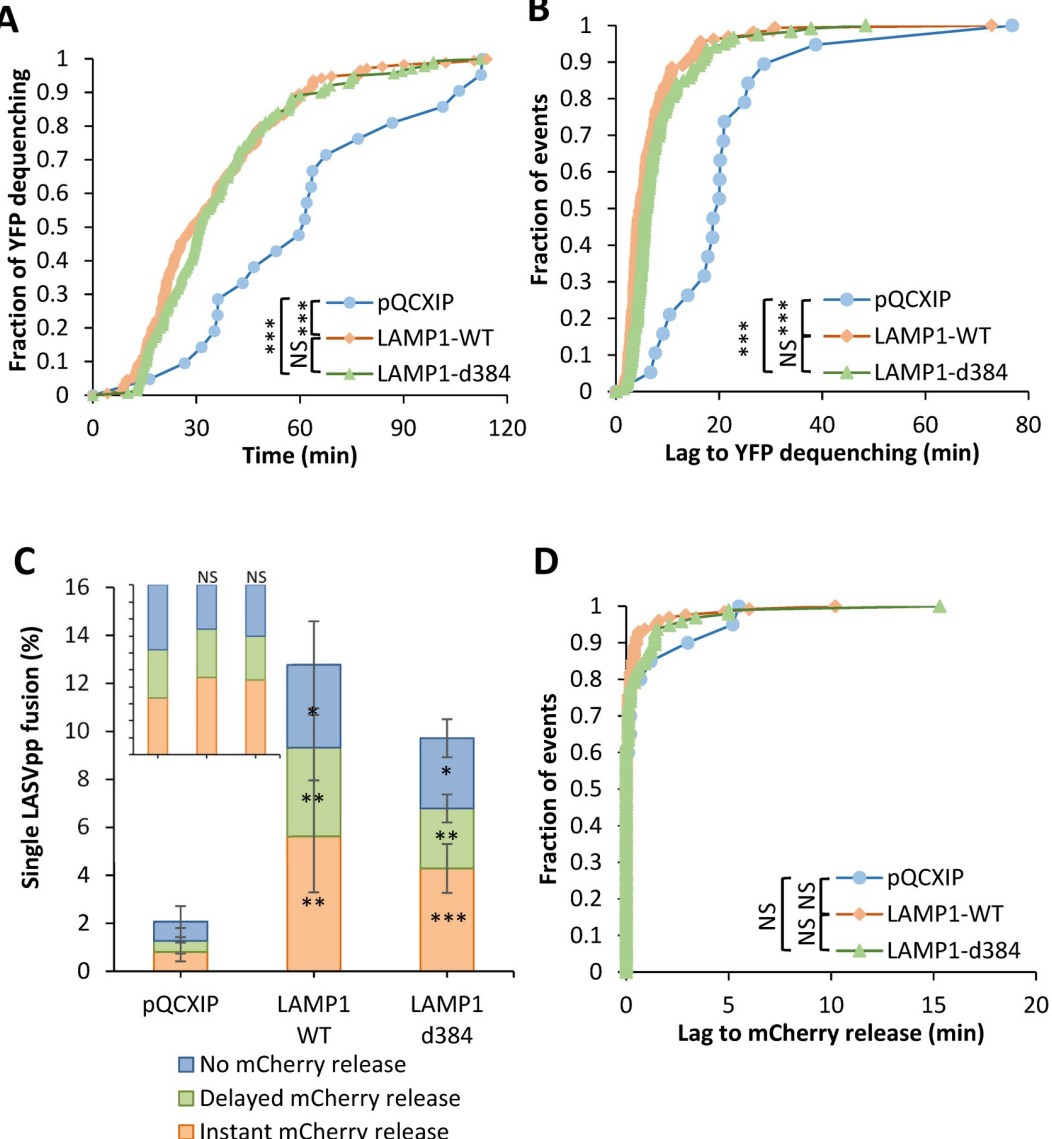

**Fig 5. Human LAMP1 expression promotes single LASVpp fusion with DF-1 cells. (A)** Efficiencies of single LASVpp fusion with instant mCherry release, delayed mCherry release, and without mCherry release in pQCXIP, LAMP1-WT and LAMP1-d384 DF-1 cells. Data shown are means ± SD of 5 independent experiments. Asterisks inside the bars represent significance relative to the vector control. Differences between LAMP1-WT and LAMP1-d384 are not statistically significant for all three categories of fusion. *Inset*: normalized fractions of each category of fusion. **(B)** The distribution of lag times between small fusion pore formation (YFP dequenching) and pore enlargement (loss of mCherry) for LASVpp fusion with DF-1 pQCXIP, LAMP1-WT and LAMP1-d384 cells. **(C)** Kinetics of small pore formation (YFP dequenching) for single LASVpp fusion events in control and hLAMP1 expressing DF-1 cells. **(D)** The distribution of lag times between LASVpp membrane permeabilization (YFP quenching) and small fusion pore formation (YFP dequenching) for LASVpp fusion with DF-1 pQCXIP, LAMP1-WT and LAMP1-d384 cells. Normalized fractions of different single virus fusion events were analyzed by Fisher's exact test using R Project. Data of lag time between YFP dequenching and mCherry release was analyzed by non-parametric Mann-Whitney test using GraphPad. Other results were analyzed by Student's t-test. *, p<0.05; **, p<0.01; ***, p<0.001; NS, not significant.

were prebound to DF-1 cells in the cold and their fusion was triggered at 37˚C by applying pH 5.0 buffer. Low pH promoted efficient single LASVpp fusion with control and hLAMP1 expressing cells (exemplified in Fig 6B and 6C). Notably, all LASVpp underwent type II fusion,

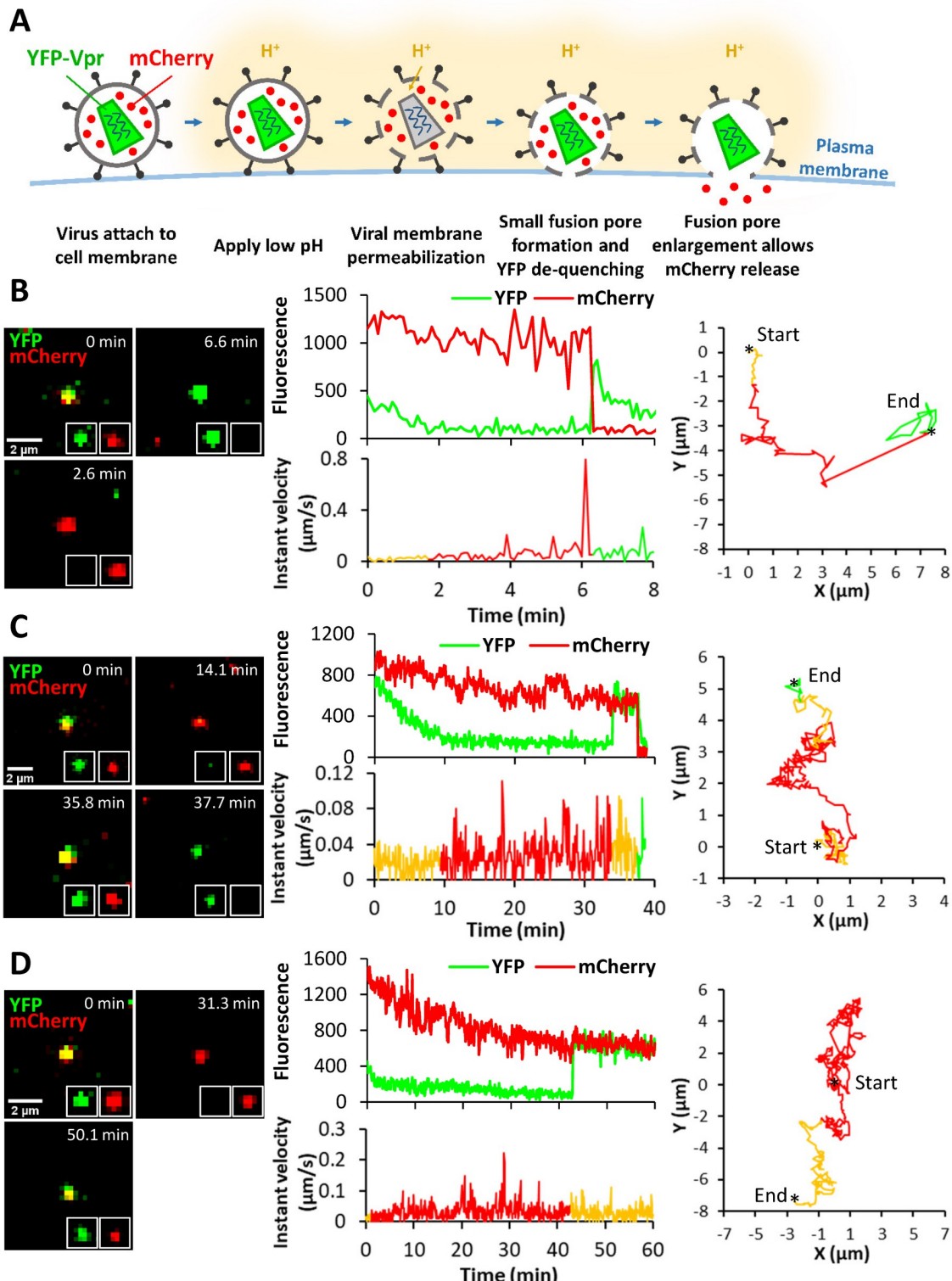

**Fig 6. Low pH-forced fusion of single LASVpp with the DF-1 cell plasma membrane.** (**A**) Illustration of single mCherry-2xCL-YFP-Vpr labeled LASVpp fusion with the plasma membrane of DF-1 cell induced by exposure to low pH. Viral membrane permeabilization in acidic buffer leads to virus interior acidification and quenching of YFP signal. Subsequent formation of a small fusion pore with the plasma membrane results in virus interior re-neutralization and YFP dequenching. Fusion pore enlargement leads to mCherry release into the cytoplasm. (**B**) Low pH-forced single LASVpp fusion with instant mCherry release. Time lapse

images (left), fluorescence traces (middle top), instant velocity (middle bottom) and trajectory (right) of single LASVpp forced fusion in DF-1-LAMP1-d384 cell showing YFP de-quenching and mCherry loss at 6.3 min. (see S4 Movie). **(C)** Low pH-forced single LASVpp fusion with delayed mCherry release relative to YFP dequenching. Time lapse images (left), fluorescence traces (middle top), instant velocity (middle bottom) and trajectory (right) of single LASVpp forced fusion with DF-1-pQCXIP cell resulting in YFP dequenching at 34 min and a subsequent loss of mCherry at 37.6 min. (see S5 Movie). **(D)** Low pH-forced single LASVpp fusion without mCherry release. Time lapse images (left), fluorescence traces (middle top), instant velocity (middle bottom) and trajectory (right) of single LASVpp fusion with a DF-1-pQCXIP cell showing YFP dequenching at 42.9 min without mCherry loss. (see S6 Movie).

similar to LASVpp fusion with endosomes. Thus, irrespective of hLAMP1 expression and the virus entry sites, LASV GPC mediates type II fusion that proceeds through permeabilization of the viral membrane prior to fusion pore opening. Similar to endosomal entry, we observed both instantaneous and delayed release of mCherry following YFP-dequenching (Fig 6B and 6C), as well as YFP dequenching without mCherry release (Fig 6D) deemed as stalled fusion events. mCherry loss from LASVpp was not a result of virus lysis, since these events should not be associated with recovery of YFP signal at low external pH. It should be stressed, however, that, although LASVpp fusion is triggered at the plasma membrane and endosome acidification is blocked by BafA1, pseudoviruses exposed to acidic pH on the cell surface can still be engulfed by and fuse with endosomes equilibrated with external acidic pH. In fact, the trajectories of some particles traveling several microns, sometimes in directional manner and with velocity reaching 0.8 μm/sec (Fig 6B–6D), are more consistent with endosomal transport of viruses prior to fusion than with viruses stuck at the cell surface.

As expected for forced fusion that bypasses the need for virus uptake and delivery into acidic endosomes, the kinetic of forced fusion with DF-1 cells was considerably faster than the kinetic of endosomal fusion (Figs 5A and 7A). Importantly, forced fusion with hLAMP1 expressing cells was markedly (~5-fold) faster than forced fusion with control cells (Fig 7A). Furthermore, the extent of forced LASVpp fusion with control DF-1 cells was approximately an order of magnitude higher than fusion through an endosomal route (compare Figs 7B and 5C). The lower efficiency of endosomal fusion might be due to a limited fraction of cell-bound LASVpp being internalized and delivered into acidic endosomes (S8A Fig), whereas all surface exposed pseudoviruses are synchronously triggered through the forced fusion protocol. In sharp contrast to a dramatic increase in the extent of forced fusion measured by the BlaM assay (Fig 2C), hLAMP1 expression did not strongly enhance the already efficient formation of small pores (detected by YFP dequenching) induced by forced fusion of single LASVpp (Fig 7B). However, fusion events associated with simultaneous YFP dequenching and mCherry release were strongly enhanced by hLAMP1 expression as compared to other types of single LASVpp fusion events (Fig 7B, *Inset*). These findings suggest that hLAMP1 promotes the initial dilation of nascent LASVpp fusion pores triggered at the cell surface to sizes that allow mCherry release. The overall fraction of single LASVpp fusion events associated with mCherry release (delayed or instantaneous) reached ~25% of all cell-bound particles for DF-1 cells expressing LAMP1-d384 (Fig 7B).

We further assessed the ability of hLAMP1 to promote fusion pore dilation by analyzing the lag time between YFP dequenching (nascent pore formation) and mCherry release as a metric for initial enlargement of fusion pores. hLAMP1 expression caused a highly significant decrease in the lag time between YFP-dequenching and mCherry release (Fig 7C). The average lag time was reduced from 3.9±0.7 min in control cells to 1.2±0.2 and 0.6±0.2 min in WT and mutant hLAMP1 expressing cells, respectively (Fig 7C), in excellent agreement with the greater fraction of "instantly" dilating pores (Fig 7B, *Inset*).

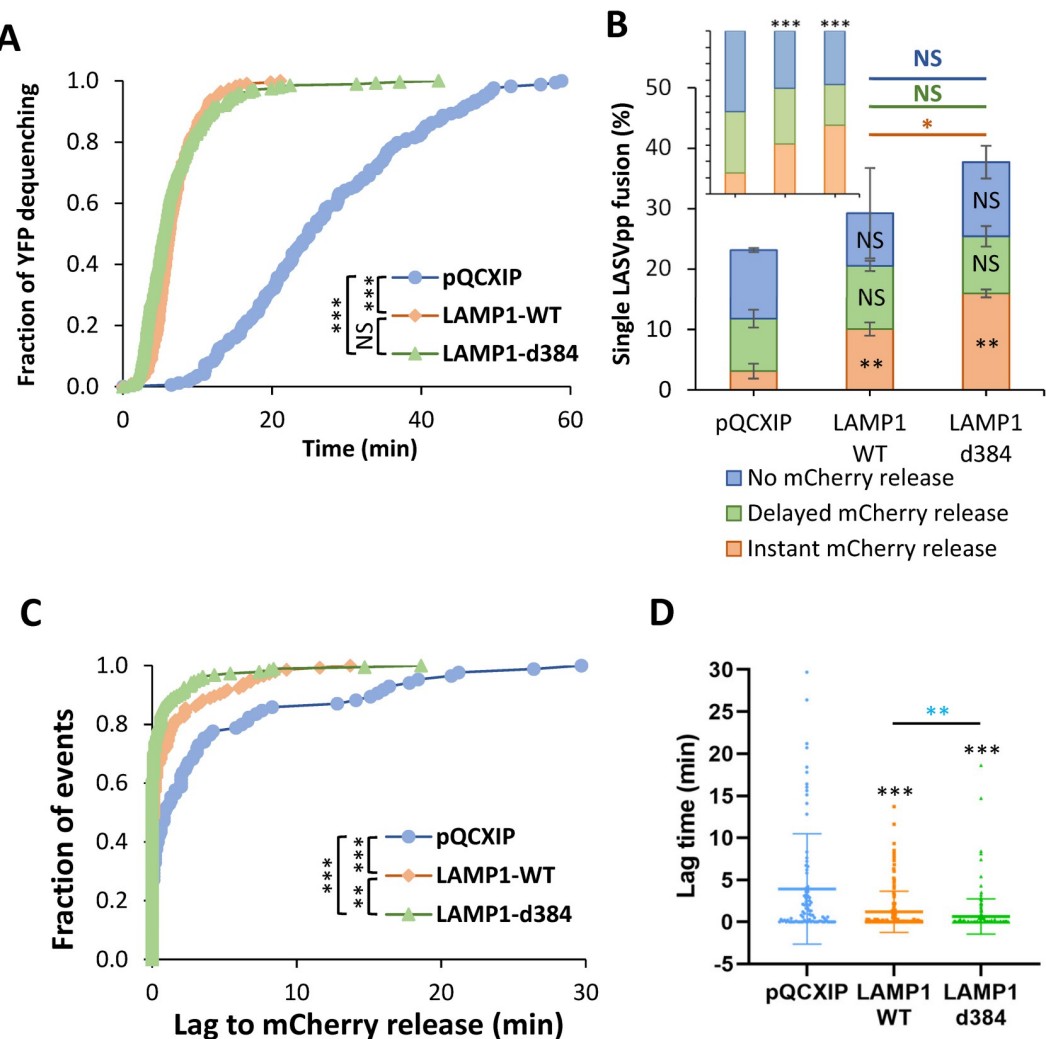

**Fig 7. Human LAMP1 enhances low pH-forced fusion of LASVpp with DF-1 cells and dramatically increases the fusion kinetics. (A)** Efficiencies of low pH-forced single LASVpp fusion events with instant mCherry release, delayed mCherry release and without mCherry release (fusion pore dilation) with DF-1 pQCXIP, LAMP1-WT and LAMP1-d384 cells. Data shown are means ± SD of 2 independent experiments. Black asterisks inside the bars represent significance relative to the vector control. The red, green and blue asterisks show significance levels for the difference between LAMP1-WT and LAMP1-d384 for fusion events with instant mCherry release, delayed mCherry release and without mCherry release respectively. At least 500 cell-bound particles were analyzed for each condition. *Inset*: normalized fractions of each category of fusion. **(B)** The distribution of lag times between small fusion pore formation (YFP dequenching) and pore enlargement (loss of mCherry) for LASVpp fusion with DF-1 pQCXIP, LAMP1-WT and LAMP1-d384 cells. **(C)** Same as in B but plotted as a dot-plot (lines represents means and whiskers are standard deviations). **(D)** The kinetics of small pore formation (YFP dequenching) between single LASVpp and DF-1 pQCXIP, LAMP1-WT and LAMP1-d384 cells using a forced fusion protocol. Data in (A) and (C) are means ± SD of two independent experiments. Blue asterisks show the significance levels for the difference between LAMP1-WT and LAMP1-d384, black asterisks on the top of bars represent significance relative to the vector control. Normalized fractions of different single virus fusion events were analyzed by Fisher's exact test using R Project. Data of lag time between YFP dequenching and mCherry release was analyzed by non-parametric Mann-Whitney test using GraphPad. Other results were analyzed by Student's t-test. *, p<0.05; **, p<0.01; ***, p<0.001; NS, not significant.

The faster kinetics of fusion and significant reduction in the lag time between YFP dequenching and mCherry loss show that hLAMP1 accelerates the forced LASVpp fusion and promotes the initial enlargement of nascent pores. The hLAMP1 effect on small pore enlargement in these experiments contrasts with a non-significant effect on enlargement of LASVpp

pores formed in endosomes (Fig 5C and 5D). Note, however, that the fraction of "instantly" dilating pores was twice as high upon endosomal fusion relative to the forced fusion (*insets* to Figs 5C and 7B), indicating a more efficient enlargement of LASVpp fusion pores formed in endosomes.

## The transmembrane domain of hLAMP1 may be required for efficient dilation of LASVpp fusion pores

To determine if hLAMP1 must be anchored to a cell membrane to facilitate LASVpp fusion, we expressed and purified the human LAMP1 ectodomain (referred to as soluble LAMP1, sLAMP1) in Expi293F cells (S9 Fig) and assessed its effect on forced LASVpp fusion with DF-1 cells, using BSA as a negative control. Pseudoviruses were pre-bound to DF-1 cells in the cold, and virus-cell fusion was triggered by applying a pre-warmed pH 5.0 buffer lacking or containing sLAMP1, followed by incubation at 37˚C. Forced LASV fusion in the presence of sLAMP1 primarily exhibited delayed release of mCherry after YFP dequenching or no mCherry release (stalled fusion) (Fig 8A–8D), whereas "instant" pore dilation events were relatively rare. mCherry release was abrogated in the presence of the fusion inhibitor ST-193 [22] (S10 Fig), demonstrating that these were *bona fide* viral fusion events.

Similar to the effect of full-length hLAMP1 expression, the kinetic of forced fusion was markedly accelerated by sLAMP1 compared to the BSA control (Fig 9A). However, forced LASVpp fusion in the presence of sLAMP1 was more than 2-fold slower than forced fusion with hLAMP1-expressing DF-1 cells (Figs 7A and 9A).

Soluble LAMP1 did not significantly increase the overall extent of LASVpp fusion (Fig 9B), but the fraction of "instant" mCherry release events was slightly but significantly increased (Fig 9B, *Inset*). A shorter lag between YFP dequenching and mCherry release (Fig 9C) further supports the notion that sLAMP1 can promote the initial pore dilation, albeit after a measurable delay (Fig 9B). The average lag time between YFP dequenching and mCherry release decreased from 6.7±1.0 min in control cells to 2.3±0.7 min in the presence of sLAMP1. The lower relative weight of "instantly" dilating fusion pores and the larger fraction of stalled fusion events (~30% *vs* ~60%) for forced fusion with sLAMP1-treated DF-1 cells compared to the full-length hLAMP1 expressing cells (Figs 7B vs 9B, *Insets*) indicates that the transmembrane domain of hLAMP1 may be required for efficient pore enlargement.

The marginal effect of sLAMP1 on the initial dilation of forced fusion pores at the cell surface prompted us to test whether the soluble receptor promotes bulk LASVpp fusion, using the BlaM assay. Pseudoviruses were bound to DF-1 cells in the cold and their fusion with the plasma membrane was triggered by exposing to pH 5.0 in the presence of sLAMP1. Surprisingly, soluble LAMP1 modestly reduced the bulk fusion efficiency with mock-transduced DF-1 cells and had no effect on forced fusion with DF-1 cells expressing LAMP1-WT or LAMP1-d384 (Fig 9D). This finding implies that, while sLAMP1 accelerates the formation of nascent GPC-mediated fusion pores and marginally increases the probability of initial pore dilation, it fails to promote the formation of fully enlarged fusion pores that can permit the release of the BlaM-Vpr labeled viral cores (S1A Fig) into the cytoplasm.

## Discussion

Here, we examined the role of human LAMP1 in LASV fusion with human and avian cells. A combination of single-virus tracking in live cells with functional assays monitoring productive virus fusion/infection enabled monitoring the evolution of fusion pores, from nascent pore opening to dilation to functionally relevant sizes. We have previously shown that single LASVpp undergoes type II fusion with endosomes of A549 cells, which entails viral membrane

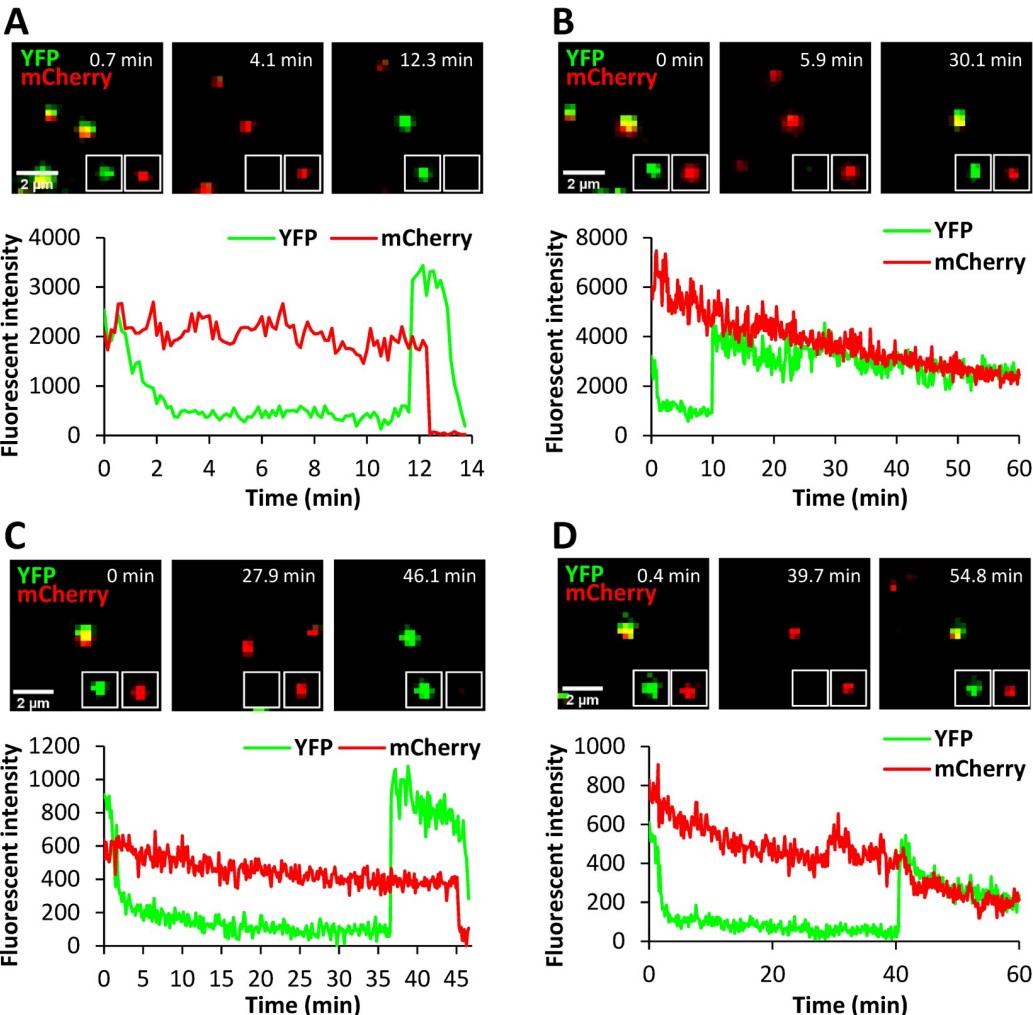

**Fig 8. Low pH-forced single LASVpp fusion with DF-1 cells in the presence of soluble LAMP1.** LASVpp were bound to cells in the cold, in the presence of 200 μg/ml soluble LAMP1 (sLAMP1) or BSA (control). Single LASVpp fusion with the plasma membrane was initiated by addition of 2 ml of warm pH 5.0 citrate buffer supplemented with 200 μg/ml sLAMP1 or BSA. **(A)** Time lapse images (top) and fluorescence traces (bottom) for single LASVpp fusion with DF-1 cell in the presence of sLAMP1 showing YFP dequenching at 11.7 min and mCherry loss at 12.7 min. **(B)** Time lapse images (top) and fluorescence traces (bottom) of single LASVpp fusion with DF-1 cell in the presence of sLAMP1 showing YFP dequenching at 11.7 min without mCherry loss. **(C)** Time lapse images (top) and fluorescence traces (bottom) of single LASVpp fusion with DF-1 cell in the presence of BSA (control) showing YFP dequenching at 36.4 min and mCherry loss at 46 min. **(D)** Same conditions as in (A), but small fusion pore formation (YFP dequenching) at 40.1 min does not culminate in mCherry release.

permeabilization (YFP-Vpr signal quenching in acidic endosomes) prior to fusion (YFP dequenching and release of the mCherry content marker) [20]. Type II fusion appears unique to LASVpp GPC, since HIV-1 particles pseudotyped with several other viral fusion proteins undergo type I fusion without prior viral membrane permeabilization [23–30]. Here, we show that LASVpp undergoes type II fusion independent of a target cell (A549 and DF-1 cells) or expression of hLAMP1, and irrespective of whether fusion occurred in endosomes or at the plasma membrane. The mechanism by which LASV GPC increases the permeability of the viral membrane upon exposure to low pH is currently under investigation.

Unexpectedly, ectopic expression of hLAMP1 has a relatively modest effect on LASV fusion/infection with endosomes across three platforms–LASVpp, LASV-VLP and

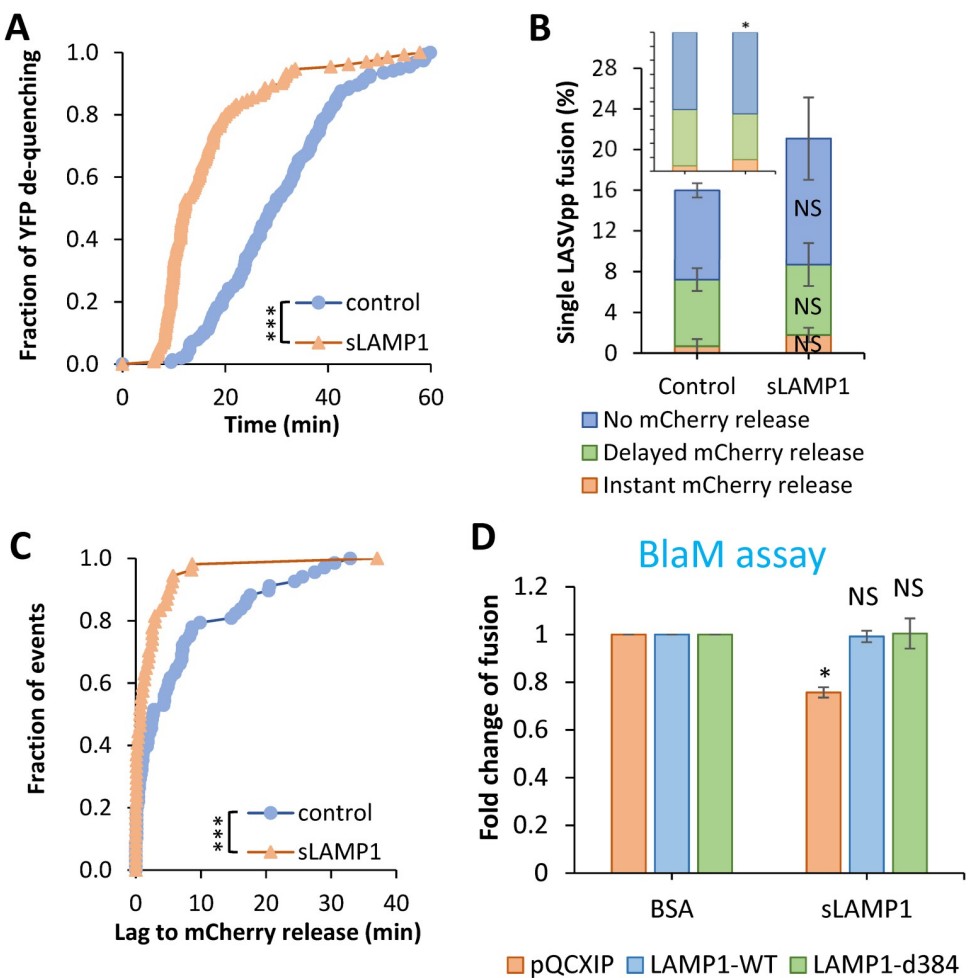

**Fig 9. Soluble LAMP1 accelerates low pH-forced single LASVpp fusion with DF-1 cells. (A)** Efficiencies of low pH-forced single LASVpp fusion events with instant mCherry release, delayed mCherry release and without mCherry release with DF-1 cells in the absence or presence of sLAMP1. Data shown are means ± SD of two independent experiments. *Inset*: normalized fractions of each category of fusion. Asterisk shows significant change relative to the BSA control. **(B)** The distribution of lag times between small fusion pore formation (YFP dequenching) and fusion (loss of mCherry) for forced LASVpp fusion with DF-1 cells in the absence or presence of sLAMP1. **(C)** Kinetics of forced single LASVpp fusion pore formation (YFP dequenching) with DF-1 cells in the absence or presence of sLAMP1. **(D)** Low pH-forced fusion of LASVpp with DF-1 cells in the presence of sLAMP1 measured by the BlaM assay. Cells were pretreated with 0.2 μM BafA1 for 1 h prior to binding pseudoviruses in the cold. Fusion was triggered by applying pH 5.0 citrate buffer at 37˚C for 20 min followed by additional incubation at a neutral pH, 37˚C for 30 min. sLAMP1 (200 μg/ml) or BSA were included throughout virus spinoculation onto cells and low pH triggering of fusion. Data shown are means ± SD of two independent experiments. Normalized fractions of different single virus fusion events were analyzed by Fisher's exact test using R Project. Data of lag time between YFP dequenching and mCherry release were analyzed by non-parametric Mann-Whitney test using GraphPad. Other results were analyzed by Student's t-test. *, p<0.05; ***, p<0.001; NS, not significant.

recombinant LCMV/LASV-GPC viruses, even in avian cells expressing a LAMP1 ortholog that does not support LASV fusion. These results indicate that productive LASVpp fusion with endosomes may be controlled by additional endosomal host factors. Indeed, we have recently shown that the late endosome-resident lipid, bis(monoacylglycero)phosphate (BMP), greatly promotes the dilation of LASV GPC-mediated fusion pores [12].

In sharp contrast to a modest effect of hLAMP1 expression on LASV entry through an endosomal pathway, we observed a dramatic enhancement of bulk LASV fusion/infection by

ectopically expressed hLAMP1 upon forcing fusion at the surface of A549 and DF-1 cells (Figs 2 and S2B). This striking difference in hLAMP1-dependence of fusion between the conventional and forced LASV entry pathways is suggestive of key differences in the requirements for productive fusion with the plasma membrane *vs* endosomes. Clearly, hLAMP1 plays a critical role in LASV fusion with the plasma membrane which likely lacks requisite endosomal co-factors that augment LASV fusion. As proposed above, the effect of hLAMP1 on LASV-endosome fusion may be masked by additional endosomal factors promoting productive LASVpp fusion and infection. Besides BMP, which we have shown to promote the enlargement of arenavirus GPC-mediated fusion pores [12], LASV fusion may be promoted by the voltage-gated calcium channels [31]. Interestingly, forced LCMV-LASV GPC infection was much less dependent on hLAMP1 expression than forced LASVpp infection (Figs 2E and 3C). This difference may reflect a more efficient dilation of fusion pores in the context of a recombinant LCMV/LASV virus, perhaps because of a greater surface density of GPCs, as compared to HIV-based pseudoviruses.

The inefficient enlargement of LASVpp pores at the cell surface through forced fusion is further supported by the much more potent promotion of the BlaM signal and infection upon hLAMP1 expression in DF-1 cells compared to a modest increase of forced single virus fusion (Figs 2C and 2E *vs* Fig). This differential effect is likely due to distinct requirements for the functional pore dilation for the bulk BlaM/infectivity assays compared to a single-virus fusion assay. It should be stressed that the latter assay cannot monitor fusion pore dilation beyond sizes that allow mCherry release; once mCherry is released, there is no way of knowing whether a fusion pore further expanded or remained small. For this reason, single virus imaging reports the formation and initial dilation of nascent fusion pores and not the formation of fully enlarged pores that can result in infection. While single virus tracking enables highly sensitive detection of nascent sub-nanometer/nanometer fusion pores, the BlaM signal and productive infection likely require full dilation of pores to release the HIV-1 capsid core (~100 nm [32]) into the cytoplasm. It should be noted that a previous study [33] has detected inadvertent cleavage of a BlaM-Vpr construct incorporated into HIV-1 virions by the viral protease and concluded that free BlaM molecules (which can diffuse through a small fusion pore) are responsible for cleavage of a BlaM substrate in infected cells, which is a readout for fusion. However, due to the lack of detectable free BlaM in our virus samples (S1A Fig), it is more likely that BlaM signal originates from the HIV-1 core-incorporated BlaM-Vpr, which is released into the cytoplasm through a fully enlarged fusion pore. This notion is further supported by a marked hLAMP1-mediated enhancement of the BlaM signal upon forced fusion of LASV-VLPs containing NP-BlaM chimera, which is not cleaved inside VLPs (S1B Fig). This result further suggests that the hLAMP1 effect on dilation of LASV GPC-mediated fusion pores is independent of the virus context (HIV-1 core *vs* arenavirus VLP).

Besides the effect of hLAMP1 on fusion pore dilation, overexpression of this intracellular LASV receptor strongly accelerates GPC-mediated fusion with endosomes and at the cell surface (through forcing fusion by low pH exposure). The faster kinetics of LASVpp fusion with endosomes may be due to a less acidic pH threshold for fusion with LAMP1 expressing membranes [10,12] which enables virus entry from earlier endosomal compartments. It appears that GPC-hLAMP1 interactions at low pH can trigger more concerted conformational changes in the LASV fusion protein, leading to a more efficient and quick formation of functional prefusion complexes, likely consisting of several activated trimers. The fusion-accelerating effect of hLAMP1 appears independent from the transmembrane domain of LAMP1, since soluble LAMP1 also accelerates forced LASVpp fusion at the cell surface, whereas the transmembrane domain may be required for efficient pore dilation (Fig 9). Thus, the hLAMP1-mediated acceleration of nascent fusion pore opening may be distinct from its promotion of LASV pore

dilation. It is currently unclear whether the transmembrane domain of hLAMP1 has a specific role in enlargement of LASV GPC-mediated fusion pores, or it simply anchors the ectodomain to a target membrane, thereby increasing the local density of this receptor for more efficient triggering of GPC refolding at low pH.

To reconcile the above observations, we propose the following model for LASV fusion. LASV is capable of forming small pores with endosomes of less permissive DF-1 cells and with the plasma membrane upon low pH exposure in the absence of hLAMP1. However, the majority of LASV fusion pores, especially those formed through the forced fusion protocol, does not enlarge to sizes that allow the cytosolic delivery of BlaM-Vpr containing cores or infection. In fact, stalled fusion pores (YFP dequenching without mCherry loss) formed upon forcing virus fusion by low pH represent nearly half of events for control DF-1 cells (Fig 7A), supporting the notion of incomplete dilation of a major fraction of fusion pores. Based on the dramatic increase in the efficiency of forced fusion and infection upon ectopic hLAMP1 expression (Fig 2), this receptor profoundly increases the formation of fully expanded functional fusion pores with suboptimal targets, such as the plasma membrane. The much-accelerated LASV fusion kinetics observed in our experiments and a shift in the pH-optimum for fusion toward less acidic pH [10,12] in the presence of hLAMP1 imply that this intracellular receptor favors the initiation of LASV fusion with earlier endosomal compartments. It appears, however, that productive LASV fusion with endosomes is augmented by additional host factors, such as BMP, that promote the dilation of LASV fusion pores [12]. Considering the difficulties associated with controlling the conditions (e.g., pH and lipid composition) in endosomes and dissecting the roles of endosomal co-factors in viral fusion, redirecting the viral fusion to a less optimal target membrane, e.g., cell plasma membrane, is a viable strategy to unraveling the effects of host co-factors on productive virus fusion.

## Materials and methods

### Cell lines and transfection

Human embryonic kidney 293T/17 cells, human lung epithelial A549 cells and chicken embryonic fibroblast DF-1 cells were obtained from ATCC (Manassas, VA, USA). A549 and DF-1 cells stably expressing wild-type and the Ala384-deleted LAMP1 mutant (LAMP1-WT and LAMP1-d384) were generated by transduction with pQCXIP retroviral vectors (Clontech, Mountain View, CA) encoding LAMP1-WT and LAMP1-d384. pQCXIP empty vector was used to generate control A549 and DF-1 vector cell lines. NuExpi293F cells were a gift of Dr. Jens Wrammert (Emory University).

Except for Expi293F cell, cells were maintained in high glucose Dulbecco's Modified Eagle Medium (DMEM; Mediatech, Masassas, VA, USA) containing 10% heat-inactivated Fetal Bovine Serum (FBS; Atlanta Biologicals, Flowery Branch, GA, USA) and 1% penicillin/streptomycin (GeminiBio, West Sacramento, CA, USA). For HEK 293T/17 cells, the growth medium was supplemented with 0.5 mg/ml G418 (Genesee Scientific, San Diego, CA, USA). Expi293 cells were maintained in Expi293 Expression Medium (Life Technologies Corporation, NY, USA).

HEK293T/17 cells were transfected with JetPRIME transfection reagent (Polyplus-transfection, Illkirch-Graffenstaden, France) according to the manufacturer's instructions. Expi293F cells were transfected with Sinofection Transfection Reagent (SinoBiological, Beijing, P.R. China) according to the manufacturer's instructions.

### Cloning and stable cell lines construction

To generate the plasmids pQCXIP-LAMP1-WT and pQCXIP-LAMP1-d384, LAMP1-WT and LAMP1-d384 segments were amplified from pcDNA-LAMP1-WT and pcDNA-LAMP1-d384

(a gift of Dr. Ron Diskin, Weizmann Institute) respectively by PCR (forward primer: GCACCGGTATGGCGGCCCCCGGCAGCGC; reverse primer for LAMP1 WT: CGCGGATCCCTAGATAGTCTGGTAGCCTGCG; reverse primer for LAMP1-d384: CGCGGATCCCTAGATAGTCTGGTAGCCGTG) and inserted into pQCXIP by *AgeI*/ *BamHI* restriction enzyme digestion and ligation.

JUNV-NP and JUNV-Z plasmids were a gift of Dr. Jack H. Nunberg (University of Montana). To generate the plasmid JUNV-NP-BlaM, β-lactamase segments were amplified by PCR (forward primer: CCATGAGGAGTGTTCAACGAAACACAGTTTTCAAGGTGGGAA GCTCCGGCGACCCAGAAACGCTGGTGAAAG; reverse primer: GGTCAGACGCCA ACTCCATCAGTTCATCCCTCCCCAGGCCGGAGCTGCCCCAATGCTTAATCAGTG AGGCACC) and inserted into JUNV-NP at NP amino acid residue 93 by QuickChange PCR with the β-lactamase segments served as megaprimers.

To create pQCXIP vector cells and cells stably expressing LAMP1-WT and LAMP1-d384, pseudotyped retroviruses were produced by transfecting HEK293T/17 cells with pQCXIP/ pQCXIP-LAMP1-WT/pQCXIP-LAMP1-d384, MLV-Gag-Pol and VSV-G using JetPRIME transfection reagent. Supernatants were harvested 36–48 h post-transfection and filtered with 0.45 μm filter to remove cell debris and virus aggregates. A549 or DF-1 cells were infected with the pseudoviruses and selected in a growth medium containing 1.5 or 2.5 μg/ml puromycin at 24 h after infection for A549 and DF-1 cells, respectively.

## Immunostaining for LAMP1

To confirm the expression of LAMP1, cells were seeded on 8-well chambered coverslips (Lab-Tek, MA, USA). Cells were washed with $PBS^{++}$, fixed with 4% PFA (Electron Microscopy Sciences, PA, USA) at room temperature for 15 min, permeabilized with 125 μg/ml digitonin (Research Products International, IL, USA) at room temperature for 15 min, and incubated with 10 μg/ml of mouse anti-hLAMP1 H4A3 antibody (Abcam Inc, Waltham, MA, USA) at room temperature for 1 hour. To assess the LAMP1 expression level on the cell surface, cells were incubated with anti-hLAMP1 antibody in the cold before PFA fixation, without cell permeabilization. Cells were then washed and incubated with 1 μg/ml AlexaFluor647 Donkey Anti-Mouse IgG (H+L) (Thermo Fisher Scientific Corporation, OR, USA) at room temperature for 45 min. Cell nuclei were stained with 10 μM Hoechst-33342 (Molecular Probes, OR, USA). Images were acquired on a Zeiss LSM 880 confocal microscope using a plan-apochromat 63X/1.4NA oil objective. LAMP1 expression level were quantified by integrating the fluorescence intensity per cell after background subtraction using ImageJ.

## Soluble LAMP1 expression and purification

Expi293F cells were transfected with pHLsec-Lamp1 fragment, a kind gift of Juha T. Huiskonen (University of Oxford, Oxford, UK), followed by three days incubation at 37˚C in the presence of 1 μg/ml of kifunensine (R&D Systems, MN, USA). Supernatant was collected and combined with half volumes of the supernatant of binding buffer (25 mM HEPES, 150 mM NaCl, pH 7.2). Soluble LAMP1 fragment (sLAMP1) was purified by Ni-NTA affinity chromatography. The elution was desalted and diluted in PBS and concentrated to 5 mg/ml. Purity of sLAMP1 was assesses by SDS-PAGE and Coomassie Blue staining and Western-blotting.

## Pseudovirus and VLP production

HEK293T/17 cells were seeded in growth medium supplemented with 10% FBS a day before transfection. For the bulk virus-cell fusion assay, LASVpp or VSVpp carrying β-lactamase-Vpr chimera (BlaM-Vpr) were produced by transfecting HEK293T/17 cells with Lassa GPC or

VSVpp, pR9ΔEnv, BlaM-Vpr and pcRev plasmids. LASV-VLPs carrying β-lactamase-NP (JUNV candid-1) chimera (JUNV-BlaM-NP) were produced by transfecting HEK293T/17 cells with Lassa GPC, JUNV-NP, JUNV-BlaM-NP and JUNV-Z plasmids. For single virus fusion experiments, dual-labeled LASVpp were produced by transfecting HEK293T/17 cells with Lassa GPC, pR9ΔEnv, mCherry-2xCL-YFP-Vpr [19] and pcRev plasmids. Supernatants were collected at 36–48 h post-transfection, filtered with 0.45 μm filter to remove cell debris and virus aggregates. LASVpp-BlaM virus was concentrated 10 times with Lenti-X concentrator (Clontech Laboratories, Mountain View, CA, USA) according to the manufacturer's instructions. The β-lactamase in LASVpp-BlaM and LASV-VLP-BlaM incorporation and cleavage were examined by Western blotting using anti-β-lactamase antibody (QED Bioscience Inc, CA, USA).

### Virus-cell fusion assay

Target cells were seeded in phenol red-free DMEM supplemented with 10% FBS. LASVpp-BlaM, VSVpp-BlaM or LASV-VLP-BlaM particles were bound to cells by centrifugation at 4°C for 30 min at 1550xg. Cells were washed with cold phenol red-free DMEM/10% FBS buffered by 20 mM HEPES (GE Healthcare Life Sciences) to remove unbound viruses. Viral fusion was initiated by shifting to 37°C for 2 h, after which time, cells were placed on ice, loaded with the CCF4-AM substrate (Life Technologies), and incubated overnight at 11°C. In control experiments, fusion was performed in the presence of 40 mM $NH_4Cl$ to raise endosomal pH and block virus entry. The cytoplasmic BlaM activity (ratio of blue to green fluorescence) was measured using a SpectraMaxi3 fluorescence plate reader (Molecular Devices, Sunnyvale, CA, USA). The background BlaM signal (blue/green ratio) in the presence of $NH_4Cl$ was subtracted from the fusion signal before calculating the fold-increase in fusion upon hLAMP1 expression. Note that, although the LASVpp infection of DF-1 cells was markedly lower than of A549 cells, DF-1 cells produced more robust BlaM signals for the same MOI. This is likely caused by better loading and/or retention of the CCF4-AM substrate by DF-1 cells.

For low pH-forced fusion at the plasma membrane, cells were pre-treated with 0.2 μM Bafilomycin A1 for 1 h. LASVpp-BlaM or LASV-VLP-BlaM particles were bound to A549 and DF-1 cells in the cold. Viral fusion was triggered by incubating the cells with citrate pH 5.0 buffer (50 mM citrate buffer, 5 mM KCl, 2 mM $CaCl_2$, 90 mM NaCl, pH 5.0, 300 mOsM) for 20 min and further incubated in phenol red-free DMEM/10% FBS for 30 min at 37°C. To assess the effect of sLAMP1 on forced LASVpp fusion, 200 μg/ml of sLAMP1 or BSA (control) was included throughout virus spinoculation onto cells and low pH triggering of fusion. The resulting BlaM activity was measured, as described above for the conventional viral entry protocol.

To avoid saturation of the BlaM signal and to make sure it is in the lineage range, different dilutions of virus stock (MOIs) were used under different conditions. For LASVpp-BlaM fusion through endosomal pathway, MOI of 0.1 and 0.05 were used for A549 and DF-1 cell lines, respectively. For the LASVpp-BlaM forced fusion, we used MOI of 5, 0.5 and 0.05 for A549-pQCXIP, DF-1-pQCXIP and for A549 or DF-1 cells expressing LAMP1-WT or LAMP1-d384, respectively. Since the titer for LASV-VLP-BlaM particles could not be determined, we used the following dilutions of the virus stock to infect different cell lines. For LASV-VLP-BlaM entry through endosomal fusion, 2x and 3x less VLPs were used to infect hLAMP1-expressing cells than control A549 and DF-1 cells, respectively. For LASV-VLP-BlaM entry through a forced pathway, 3x and 50x less VLP was used to infect LAMP1 expressing cells than control A549 and DF-1 cells, respectively.

### Infectivity assay for LCMV/LASV-GPC recombinant virus

Target cells were seeded in DMEM supplemented with 10% FBS and grown to 70% confluency. LCMV/LASV-GPC viruses (MOI 0.01) were bound to cells by centrifugation at 4˚C for 30 min at 1550xg. Cells were washed with cold phenol red-free DMEM/10% FBS buffered by 20 mM HEPES (GE Healthcare Life Sciences) to remove unbound viruses. Infection was allowed by incubating at 37˚C for 20 h. For the infection through forced pathway, cells were pretreated with 0.2 μM Bafilomycin A1 for 1 hour. LCMV/LASV-GPC virus (MOI 0.1) were bound to cell in the cold. Viral fusion was triggered by incubating the cells with citrate pH 5.0 buffer for 20 min at 37˚C and further cultured in phenol red-free DMEM/2% FBS for 20 h at 37˚C. Cells were washed with PBS++, fixed with 4% PFA at room temperature for 15 min, permeabilized with 0.3% TritonX-100 (Sigma, St. Louis, MO, USA) at room temperature for 15 min, and incubated with 5 μg/ml rat anti-LCMV NP antibody (Bio X Cell, Lebanon, NH, USA) at room temperature for 1 hour. Cells were then washed and incubated with 1 μg/ml Donkey anti-Rat polyclonal Dylight 488-conjugated antibody (Thermo Fisher Scientific Corporation, OR, USA) at room temperature for 45 min. Cell nuclei were stained with 10 μM Hoechst-33342 (Molecular Probes, OR, USA). Images were acquired by Cytation 3 imaging plate reader (Agilent Technologies, Santa Clara, CA, USA), cell nuclei and LCMV NP expression in infected cells were detected by Gen5 software (Agilent Technologies, Santa Clara, CA, USA). Infection was calculated as the fraction of infected cell.

### LASV pseudovirus infectivity assay

Target cell were seeded in phenol red-free DMEM supplemented with 10% FBS. Luciferase-encoding LASVpp were bound to cells by centrifugation at 4˚C for 30 min at 1550xg (MOI = 0.1 for A549 cells and MOI = 10 for DF-1 cells). Unbound virus was removed by washing with cold medium and the cells were incubated with 50 μl growth medium at 37˚C to allow infection. For the infection through forced pathway, cells were pretreated with 0.2 μM Bafilomycin A1 for 1 h. LASVpp were bound to cell in the cold (MOI = 1 for A549 cells and MOI = 10 for DF-1 cells). Viral fusion was triggered by incubating the cells with citrate pH 5.0 buffer for 20 min followed by further incubation with 50 μl growth medium at 37˚C to allow infection. Thirty-six hours post-infection, cells were incubated with 50 μl Bright-Glo luciferase substrate (Promega, Madison, WI) at room temperature for 5 min, the luciferase activity was measured by a TopCount NXT plate reader (PerkinElmer Life Sciences, Waltham, MA, USA).

### Single virus imaging in live cell

Target cells were seeded in 35 mm collagen coated glass-bottom Petri dishes (MatTek, MA, USA) in Fluorobrite DMEM (Life Technologies Corporation, NY, USA) containing 10% FBS, penicillin, streptomycin and L-glutamine 2 days before imaging. The binding of dual-labeled LASVpp to cells (MOI of 0.1) was facilitated by centrifugation at 4˚C, and cells were washed with cold PBS$^{++}$ to remove unbound virus. Virus entry and fusion were initiated by adding 2 ml of pre-warmed Fluorobrite DMEM containing 10% FBS and 20 mM HEPES immediately prior to imaging cells on a DeltaVision microscope equipped with a temperature, humidity and CO$_2$-controlled chamber for 2 h. Every 6 sec, 4 Z-stacks spaced by 1.5 μm were acquired to cover the thickness of cells using Olympus 60x UPlanFluo /1.3 NA oil objective (Olympus, Japan). For low pH-forced fusion at the plasma membrane, cells were pre-treated with 0.2 μM Bafilomycin A1 (Sigma, St. Louis, MO, USA) for 1 h. Forced fusion was initiated by adding 2 mL citrate pH 5.0 buffer with 0.2 μM Bafilomycin A1, followed by imaging for 1 h at 37˚C. To assess the effect of sLAMP1 on LASVpp forced fusion, 200 μg/ml of sLAMP1 or BSA (control) was added to virus during spinoculation onto cells and included in a citrate pH 5.0 buffer

during virus entry and imaging. In control experiments, 10 µM of ST-193 (MedChemExpress, NJ, USA) was applied to the mixture virus during spinoculation onto cells and the inhibitor and 0.2 mg/ml of sLAMP1 were included in citrate pH 5.0 buffer and maintained throughout imaging.

The acquired time-lapse Z-stack images were converted to maximum intensity projections for single particle tracking. Single fusion events were annotated using the ImageJ ROI manager tool. The times of YFP dequenching and mCherry loss, which occurred in a single image frame, were determined visually as the time of color change. Representative single virus fusion events were tracked using ICY image analysis software (icy.bioimageanalysis.org). The labeled pseudoviruses were identified by Spot Detection plugin and tracked using Spot Tracking plugin to determine the fluorescence intensity over time, particle trajectory and instant velocity.

## Statistical analysis

Data of lag time between YFP dequenching and mCherry release was analyzed by non-parametric Mann-Whitney test using GraphPad. Normalized fractions of different single virus fusion events were analyzed by Fisher's exact test using R Project. Other results were analyzed by Student's t-test using Excel. Unless stated otherwise, *, $p < 0.05$; **, $p < 0.01$; ***, $p < 0.001$; NS, not significant.

## Supporting information

**S1 Fig. BlaM-Vpr or NP-BlaM constructs are not cleaved in pseudoviruses (LASVpp-BlaM) and virus-like particles (LASV-VLP-BlaM).** BlaM-Vpr (**A**) or NP-BlaM (**B**) in LASVpp-BlaM and LASV-VLP-BlaM particles, respectively, were examined by Western blot-ting using anti-β-lactamase antibody. LASVpp carrying mCherry-YFP-Vpr and LASV-VLP-NP were used as negative controls for non-specific with signal.
(JPG)

**S2 Fig. Effect of hLAMP1 expression in A549 and DF-1 cells on LASVpp fusion and infec-tion (related to Fig 2).** Raw data from representative experiments are shown. (**A**) LASVpp-BlaM fusion with A549 and DF-1 cells. Due to the limited dynamic range of the BlaM assay, A549 and DF-1 cells were infected with MOI of 0.1 and 0.05, respectively, to ensure a linear range of the BlaM signal (blue/green fluorescence ratio). (**B**) Low pH-forced fusion of LASVpp with A549 and DF-1 cells. MOI of 5, 0.5 and 0.05 were used for A549-pQCXIP, DF-1-pQCXIP and for A549 or DF-1 cells expressing LAMP1-WT or LAMP1-d384, respectively. The plotted signal in panel (A) and (B) is corrected for the dilution factor. (**C**) LASVpp infection of A549 and DF-1 cells. MOI of 1 and 10 were used for A549 and DF-1 cell respectively. (**D**) LASVpp infection through low pH bypass protocol in A549 and DF-1 cells. MOI of 0.1 and 10 were used for A549 and DF-1 cell respectively. Data in panel (D) are plotted in logarithmic scale. Data shown are means ± SD of three technical replicates. Statistical significance was deter-mined by Student's t-test. *, $p < 0.05$; **, $p < 0.01$; ***, $p < 0.001$. Asterisks on the top of bars rep-resent significance relative to the vector control.
(JPG)

**S3 Fig. hLAMP1 expression in A549 and DF-1 cells does not strongly enhance pseudovirus fusion mediated by VSV G glycoprotein. (A)** VSVpp-BlaM pseudovirus fusion with A549 and DF-1 cells. VSVpp entry through an endosomal pathway was initiated by pre-binding pseudoviruses in the cold, shifting to 37˚C and incubating for 2 h. **(B)** Low pH-forced fusion of VSVpp with A549 and DF-1 cells. Cells were pretreated with 0.2 µM BafA1 for 1 h prior to

binding pseudoviruses in the cold. Fusion was triggered by applying pH 5.0 citrate buffer at 37˚C for 20 min followed by additional incubation at neutral pH, 37˚C for 30 min. Data are means ± SD of three technical replicates of a representative experiment. Statistical significance was determined by Student's t-test. *, p<0.05; NS, not significant. Asterisks on the top of bars represent significance relative to the vector control.
(JPG)

**S4 Fig. LAMP1 expression enhances LASV-VLP fusion with A549 and DF-1 cells. (A)** LASV-VLP-BlaM fusion with A549 and DF-1 cells. LASV-VLP entry through an endosomal pathway was initiated by pre-binding the VLP in the cold, shifting to 37˚C and incubating for 2 h. **(B)** Low pH-forced fusion of LASV-VLP with A549 and DF-1 cells. Cells were pretreated with 0.2 μM BafA1 for 1 h prior to binding the VLPs in the cold. Fusion was triggered by applying pH 5.0 citrate buffer at 37˚C for 20 min followed by additional incubation in a neutral pH medium at 37˚C for 30 min. **(C)** Representative raw data of LASV-VLP fusion with A549 and DF-1 shown in panels (A). **(D)** Representative raw data of LASV-VLP low pH-forced fusion results shown in panels (B). Due to the limited dynamic range of the BlaM assay, different dilutions of the virus stock were used to ensure a linear range. For LASV-VLP-BlaM entry through endosomal pathway, 2x and 3x less VLPs were used to infect hLAMP1-expressing cells than control A549 and DF-1 cells, respectively. For the forced fusion of LASV-VLP-BlaM, 3x and 50x less VLPs were used to infect LAMP1 expressing cells than control A549 and DF-1 cells, respectively. Data shown in panels (A) and (B) are means ± SD of three independent experiments. Data shown in panel (C) and (D) are means ± SD of three technical replicates. Data were analyzed by Student's t-test. **, p<0.01; ***, p<0.001.
(JPG)

**S5 Fig. Fusion of single LASVpp with the DF-1 cell. (A)** Time lapse images (top) and fluorescence traces (bottom) of single LASVpp fusion with a DF-1-pQCXIP cell showing YFP quenching at 29.5 min and YFP dequenching/mCherry loss at 46.5 min corresponding to virus interior acidification and fusion, respectively. **(B)** Time lapse images (top) and fluorescence traces (bottom) of single LASVpp fusion with a DF-1-LAMP1-d384 cell showing YFP quenching at 5 min and YFP dequenching/mCherry loss at 25 min, indicating virus interior acidification and fusion, respectively.
(JPG)

**S6 Fig. Saponin lysis of LASVpp labeled with mCherry-2xCL-YFP-Vpr attached to a coverslip.** Images (left panel) and quantification (right panel) of virus lysis (loss of mCherry from YFP-Vpr labeled particles) before and 10 min after application of saponin. A small fraction (~15%) of immature HIV-1 pseudoviruses retained mCherry. Data are means ± SD from 4 image fields analyzed.
(JPG)

**S7 Fig. Single LASVpp fusion with A549 cells. (A)** A representative LASVpp fusion event (YFP dequenching) with concomitant mCherry release (quick fusion pore dilation). **(B)** LASVpp fusion event with delayed mCherry release relative to YFP dequenching. **(C)** LASVpp fusion event (YFP dequenching) without mCherry release. For all panels, time lapse images (left), fluorescence intensity traces (middle top), instant velocity (middle bottom) and trajectory (right) are shown. **(D)** Fraction of LASVpp fusion events according to the pore enlargement phenotype (instant and delayed mCherry release or lack of mCherry release). Data are means ± SD of 3 independent experiments. **(E)** Kinetics of nascent pore formation (YFP dequenching) for single LASVpp fusion.
(JPG)

**S8 Fig. LAMP1 expression doesn't affect single LASVpp endocytosis and viral membrane permeabilization. (A)** Fraction of single LASVpp exhibiting YFP quenching in DF-1 pQCXIP, LAMP1-WT and LAMP1-d384 cells. Data shown are means ± SD of 5 independent experiments. **(B)** Kinetics of the YFP quenching of single LASVpp in control and hLAMP1 expressing DF-1 cells. Data were analyzed by Student's t-test. *, $p<0.05$; **, $p<0.01$; NS, not significant.
(JPG)

**S9 Fig. SDS-PAGE and Western-blot of purified sLAMP1.** Purified sLAMP1 were detected using Coomassie Blue staining (left) and verified by SDS-PAGE and Western-blotting using anti-hLAMP1 antibody (right).
(JPG)

**S10 Fig. ST-193 inhibits single LASVpp fusion forced by low-pH.** LASVpp were bound to DF-1 cells in the cold, in the presence of 200 μg/ml soluble LAMP1 with or without 10 μM of ST-193. Single LASVpp fusion with the plasma membrane was initiated by addition of 2 ml of warm pH 5.0 citrate buffer supplemented with 200 μg/ml sLAMP1 with or without 10 μM of ST-193. Graph show efficiencies of low pH-forced single LASVpp fusion events with instant mCherry release, delayed mCherry release and without mCherry release with DF-1 cells in the absence of sLAMP1 with or without ST-193. Data shown result of 1 independent experiment.
(JPG)

**S1 Movie. LASVpp fusion event with instant viral content release following YFP dequenching.** Single LASVpp exhibits YFP quenching (white arrow) at 31.3 min followed by YFP dequenching/mCherry loss at 34.7 min corresponding to virus interior acidification and fusion, respectively. The frame rate is slowed 10-fold around the YFP quenching and dequenching events (29:36–32:12 min and 33:42–35:54 min, respectively) to better illustrate these steps of viral fusion. Movie is related to Fig 4B.
(AVI)

**S2 Movie. LASVpp fusion event with delayed viral content release following YFP dequenching.** Single LASVpp exhibits YFP quenching (white arrow) at 39.7 min followed by YFP dequenching at 42.7 min and mCherry loss at 43.2 min corresponding to virus interior acidification, small fusion pore formation and fusion pore dilation, respectively. The frame rate is slowed 4-fold around the YFP quenching, dequenching and mCherry release events (42:24–46:54). Movie is related to Fig 4C.
(AVI)

**S3 Movie. LASVpp fusion events without viral content release.** Single LASVpp exhibits YFP quenching (white arrow) at 42.6 min followed by YFP dequenching at 47.4 min corresponding to virus interior acidification and small fusion pore formation, respectively. The frame rate is slowed 4-fold around the YFP quenching and dequenching events (41:24–48:24 min). Movie is related to Fig 4D.
(AVI)

**S4 Movie. Low pH-forced LASVpp fusion with instant viral content release.** Single LASVpp exhibits YFP quenching (white arrow) at 1.3 min followed by YFP dequenching/mCherry loss at 6.3 min corresponding to virus interior acidification and fusion, respectively. Movie is related to Fig 6B.
(AVI)

**S5 Movie. Low pH-forced LASVpp fusion with delayed viral content release.** Single LASVpp exhibits YFP quenching (white arrow) at 9.5 min followed by YFP dequenching at 34 min and mCherry loss at 37.6 min corresponding to virus interior acidification, small fusion pore formation and fusion pore dilation, respectively. The frame rate is slowed 10-fold around the YFP quenching and dequenching events (5:54–11:00 min and 32:54–38:24 min, respectively). Movie is related to Fig 6C.
(AVI)

**S6 Movie. Low pH-forced LASVpp fusion without viral content release.** Single LASVpp exhibits YFP quenching (white arrow) at 1.4 min followed by YFP dequenching at 42.9 min corresponding to virus interior acidification and small fusion pore formation, respectively. The frame rate is slowed 20-fold around the YFP quenching and dequenching events (0–2:00 min and 42:00–44:00 min, respectively). Movie is related to Fig 6D.
(AVI)

## Acknowledgments

The authors wish to thank Jack Nunberg (University of Montana) for the gift of JUNV-NP and JUNV-Z plasmids and cloning advice, Ron Diskin (Weizmann Institute) for WT and mutant human LAMP1 expression vectors, and Juha Huiskonen (University of Oxford) for the gift of sLAMP1 expression vector. We are also grateful to Xiangyang Guo for advice on sLAMP1 expression and purification, Baek Kim (Emory University) for access to Cytation 3 plate reader, and the members of Melikyan lab for critical reading of the manuscript and helpful comments.

## Author Contributions

**Conceptualization:** Gregory B. Melikyan.

**Data curation:** You Zhang.

**Formal analysis:** You Zhang, Gregory B. Melikyan.

**Funding acquisition:** Gregory B. Melikyan.

**Investigation:** You Zhang.

**Methodology:** Juan Carlos de la Torre, Gregory B. Melikyan.

**Project administration:** Gregory B. Melikyan.

**Resources:** Juan Carlos de la Torre, Gregory B. Melikyan.

**Supervision:** Gregory B. Melikyan.

**Visualization:** You Zhang.

**Writing – original draft:** You Zhang, Gregory B. Melikyan.

**Writing – review & editing:** You Zhang, Juan Carlos de la Torre, Gregory B. Melikyan.

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
