## [Decision Letter · Decision Letter 0]

27 Jun 2022

Dear Dr. Melikyan,

Thank you for submitting your manuscript "Human LAMP1 accelerates Lassa virus fusion and potently promotes fusion pore dilation upon forcing viral fusion with non-endosomal membrane" for consideration at PLOS Pathogens. As with all papers reviewed by the journal, your manuscript was reviewed by members of the editorial board and by several independent reviewers. The reviewers were generally enthusiastic about your work, although the extent of enthusiasm differed as you will see from the comments below.  Provided that you can satisfactorily address the comments of reviewers 2 and 3 we would be delighted to accept your manuscript for publication.

Sincerely,

Sean P.J. Whelan

Associate Editor

PLOS Pathogens

Susan Ross

Section Editor

PLOS Pathogens

Kasturi Haldar

Editor-in-Chief

PLOS Pathogens

orcid.org/0000-0001-5065-158X

Michael Malim

Editor-in-Chief

PLOS Pathogens

orcid.org/0000-0002-7699-2064

Reviewer Comments (if any, and for reference):

Reviewer's Responses to Questions

**Part I - Summary**

Reviewer #1: Zhang et al. report an observational study of the role of LAMP1 binding to GPC during LASV entry, continuing an important body of work from the Melikyan laboratory. Previous work from this group and others has demonstrated that LAMP1 is not strictly required for LASV entry but enables entry earlier in the endocytic pathway. The mechanistic basis for this role of LAMP1 is accelerating LASV entry is unclear. By redirecting LASV entry to the plasma membrane through ectopic expression of LAMP1 and exposure of cells to acidic pH, they demonstrate a previously unknown role for LAMP1 binding to GPC in accelerating fusion pore expansion. Examining the role of LAMP1 during LASV entry at the plasma membrane enabled the authors to specify the contribution of LAMP1 in the absence of other endosomal factors that might impact LASV entry (e.g. the anionic lipid BMP, which the authors previously demonstrated promotes LASV fusion).

The manuscript is very well written, and the data are clearly presented. The results reflect a meaningful contribution to the understanding of LASV entry and the role of the LAMP1 receptor. The manuscript also demonstrates the value of monitoring viral entry at the plasma membrane, although it might not be the natural site of entry. This approach will be of general interest to other groups engaged in the study of viruses that enter through the endosome.

Reviewer #2: This excellent, very well controlled paper presents in-depth analysis of the mechanisms of membrane fusion mediated by the virus glycoprotein complex (GPC) of Lassa virus (LASV) and focuses on the role of LAMP1 that is not absolutely required for fusion and infection but rather regulates the timing and efficiency of viral entry. The authors use an exceptionally wide arsenal of state-of-the-art approaches and models to explore LAMP1 contributions in different fusion stages. Most importantly, the analysis of the single-virus fusion to endosomal and plasma membranes in live cells allows very sensitive detection of pre-fusion pores in viral envelope, nascent proton-permeable fusion pores and expanding fusion pores. Among many novel features of GPC-mediated features uncovered in this work, I am most intrigued by a much stronger LAMP1 dependency of fusion to plasma membrane vs. that to endosomal membrane and by the conclusion that LAMP1 influences more than one stage in the fusion pathway. The results of this exciting work will be of interest to researchers working on the cell entry mechanisms of diverse enveloped viruses. I have only minor suggestions/questions.

Reviewer #3: In this study, Zhang et al. investigate the role of LAMP1 in supporting membrane fusion mediated by LASV GPC displayed on HIV pseudoparticles, LASV VLPs, and recombinant LASV. They employ a combination of assays measuring both bulk functional consequences of manipulations and single-particle kinetic tracking in live cells. They use two cells lines, a human cell line endogenously expressing LAMP1 that supports LASV fusion, and an avian cell line expressing a LAMP1 ortholog that does not support LASV fusion. They measure fusion in two distinct contexts, from within endosomes (endogenous pathway) or from the plasma membrane (forced pathway). To assess the contribution of LAMP1 on membrane fusion the authors overexpress WT human LAMP1 (hLAMP1) or a mutant version (d384) that preferentially traffics to the cell surface in either cell line. They discover a novel function for hLAMP1 in promoting fusion-pore dilation that might depend on the hLAMP1 transmembrane domain. This effect of hLAMP1 on LASV GPC fusion was only apparent or more pronounced on the plasma membrane (i.e. in the absence of other endosomal co-factors that contribute to the same process).

This study extends our view of viral membrane fusion dynamics and host factor roles at distinct stages of fusion. It is particularly exciting because the questions of fusion-pore expansion are difficult to dissect mechanistically - the authors overcome this hurdle by employing a very elegant assay that quantitatively reports on both small fusion-pore formation and its initial expansion, and by combining it with bulk functional assays. I look forward to seeing this work published in PloS Pathogens, but I have some major and minor points to be addressed before that point.

**Part II – Major Issues: Key Experiments Required for Acceptance**

Reviewer #1: I have no significant critiques to offer about this manuscript and support publication in its current form.

Reviewer #2: I have only minor comments/suggestions for clarification.

Reviewer #3: 1) It is not surprising that the detailed quantitative effects of different manipulations would have different outcomes for different virus systems or in different cell lines, and there are many possible reasons for that, but I find the early discussion of each of the results too speculative and distracting because it comes across inconsistent. I list some examples:

a. The first sentences of the second results section give two distinct interpretations for the small effect of hLAMP1 expression on LASVpp entry in A549 vs. DF-1 cells. In the former case it is interpreted as indication of near optimal fusion, and in DF-1 cells as an indication of limited access of LASVpp to endosomal compartments. Can the efficiency of fusion for internalized particles be measured in the two contexts? Or alternatively, could the efficiency of internalization in the two cell lines be compared (one supplement shows this for DF-1 cells, but how about in A549?). Furthermore, it was not clear what was meant by the limited access until much later in the manuscript where it was explained that it refers to the efficiency of endocytosis. An alternative would be to move any speculation to the Discussion section.

b. At the end of the second Results section the authors speculate on the reasons for the discordance between fold-enhancement of forced fusion vs. infection in hLAMP1 expressing A549 cells might be due to the less efficient infection following fusion at the plasma membrane. Why would the same effect not be observed for DF-1 cells?

2) Most fusion and infection figures report only fold-changes. To better evaluate the data, it would be important to also include absolute values perhaps as a Supplement.

3) From the presented data I am not convinced of the advantage of DF-1 cells over A549 cells for the detailed kinetic dissections of hLAMP1 effects in live-cell tracking experiments.

a. The very similar relative effect of hLAMP1 expression on endosomal fusion by LASVpp in A549 and DF-1 cells is surprising even in the context of the speculated lower internalization efficiency. Even if a much smaller fraction of total particles are internalized into DF-1 vs. A549 endosomes, going from no hLAMP1 to about endogenous levels of hLAMP1 would be expected to have a much larger fold change than in the case of going from having hLAMP1 to having more of it. The relative change in the measured function should not be dependent on the efficiency of internalization (which does not change with hLAMP1 expression).

b. Since there are no marked differences in the effects of hLAMP1 (over)expression in the two cell lines (and given the comment under ‘a’ above), I see no advantage of choosing DF-1 cells over the more native target. It would be important to corroborate the detailed mechanistic interpretations for the role of hLAMP1 in pore expansion in A549 cells. This would be helpful both in endosomes (same magnitude of effects in DF-1 and A549) and on the plasma membrane (since endogenous LAMP1 would not be present there just the same).

4) Instant velocity plots are shown in various figures but not mentioned or described in the text.

**Part III – Minor Issues: Editorial and Data Presentation Modifications**

Reviewer #1: (No Response)

Reviewer #2: 1) Can finding that hLAMP1 expression does not significantly alter the relative weights of the three types of fusion events: instant, delayed pore opening, no dilation (Fig. 5A) indicate that Lamp1 influences the stage(s) preceding the fusion pathway? For instance, can you exclude the effects of the LAMP1 expression at the surface of the cells beyond the relatively low endogenous levels of this receptor in A549 cells, Fig. 1C on cell surface binding of GPC carrying pseudoviruses?

2) I am somewhat unclear on the relative effects of overexpression of WT LAMP1 that is almost absent at the plasma membrane and LAMP1 d384 that is mostly expressed at the plasma membranes. Can you please further discuss why in Fig. 2B WT LAMP1 and LAMP1 d384 similarly (even if the difference is statistically significant) promote fusion in endosomes and at the plasma membrane?

3) I am intrigued by the finding that pseudoviruses exposed to acidic pH on the cell surface can still be internalized and fuse with neutral endosomes. Does this outcome (fusing in endosome rather than at the plasma membrane) correlate with how fast these virions are internalized? If fusion cannot happen faster than within certain time and if the virus happens to be internalized within this time, fusion will happen only in endosome.

4) To distinguish direct effects of LAMP1 (as a receptor) from indirect effects such as domain organization in PM one may ask whether LAMP1 expression influences fusion for viral fusogens that do not utilize LAMP1 as a receptor, say for HIV Env?

5) I suggest the sudden appearance of the term ‘type II fusion’ needs to be preceded by a brief explanation of the way you have classified single virus fusion events into Type 1 and 2 in Ref 20. Also is there any evidence to suggest that fusion is dependent on pre-fusion permeabilization? The authors show that LASVpp undergoes type II fusion independent of a target cell (A549 and DF-1 cells) and irrespective of whether fusion occurred in endosomes or at the plasma membrane. Can pre-fusion permeabilization be uncoupled from fusion in low pH-forced fusion model by using less acidic pH than pH 5.0.

6) Fig. 2D is referred to before Fig. 2C.

7) The sentence “We have previously shown that hLAMP1 overexpression allows LASV GPC-mediated fusion to occur at higher pH [10, 12].” should be edited, since Ref. 10 is published by another group.

Reviewer #3: 1) On page 11, second paragraph, it should be clearly stated that the single LASVpp fusion being imaged is endosomal fusion at that stage.

2) Page 12, line 14, it is not clear which possibility is argued against.

3) Page 14, line 13, it is commented that Fig. 6 shows efficient single LASpp fusion. It is not clear how a couple of particles report on efficiency.

4) Please make clear in the text when soluble LAMP1 was added to virus particles relative to the attachment step of viruses to cells.

5) Figure 2A legend, should explain that LASVpp are expressing luciferase and when luciferase signal was measured (perhaps move the intro sentences in part D to part A).

6) Fig. S6 – why does LAMP1 have ladder-like appearance?

7) In the discussion section, perhaps add more discussion of the endosomal factors that might be redundant to LAMP1.

PLOS authors have the option to publish the peer review history of their article (what does this mean?). If published, this will include your full peer review and any attached files.

Reviewer #1: No

Reviewer #2: No

Reviewer #3: No

Figure Files:

Data Requirements:

Reproducibility:

References:

---

## [Editor Report · Decision Letter 1]

1 Aug 2022

Dear Dr. Melikyan,

We are pleased to inform you that your manuscript 'Human LAMP1 accelerates Lassa virus fusion and potently promotes fusion pore dilation upon forcing viral fusion with non-endosomal membrane' has been provisionally accepted for publication in PLOS Pathogens.

Best regards,

Sean P.J. Whelan

Associate Editor

PLOS Pathogens

Susan Ross

Section Editor

PLOS Pathogens

Kasturi Haldar

Editor-in-Chief

PLOS Pathogens

orcid.org/0000-0001-5065-158X

Michael Malim

Editor-in-Chief

PLOS Pathogens

orcid.org/0000-0002-7699-2064
---

## [Editor Report · Acceptance letter]

9 Aug 2022

Dear Dr. Melikyan,

We are delighted to inform you that your manuscript, "Human LAMP1 accelerates Lassa virus fusion and potently promotes fusion pore dilation upon forcing viral fusion with non-endosomal membrane," has been formally accepted for publication in PLOS Pathogens.

Best regards,

Kasturi Haldar

Editor-in-Chief

PLOS Pathogens

orcid.org/0000-0001-5065-158X

Michael Malim

Editor-in-Chief

PLOS Pathogens

orcid.org/0000-0002-7699-2064